# Implicit Language Models are RNNs: Balancing Parallelization and Expressivity

**Mark Schöne** [* 1 2]  **Babak Rahmani** [* 2]  **Heiner Kremer** [2]  **Fabian Falck** [2]  **Hitesh Ballani** [2]  **Jannes Gladrow** [2]

## Abstract

State-space models (SSMs) and transformers dominate the language modeling landscape. However, they are constrained to a lower computational complexity than classical recurrent neural networks (RNNs), limiting their expressivity. In contrast, RNNs lack parallelization during training, raising fundamental questions about the trade off between parallelization and expressivity. We propose *implicit SSMs*, which iterate a transformation until convergence to a fixed point. Theoretically, we show that implicit SSMs implement the non-linear state-transitions of RNNs. Empirically, we find that only approximate fixed-point convergence suffices, enabling the design of a scalable training curriculum that largely retains parallelization, with full convergence required only for a small subset of tokens. Our approach demonstrates superior state-tracking capabilities on regular languages, surpassing transformers and SSMs. We further scale implicit SSMs to natural language reasoning tasks and pretraining of large-scale language models up to 1.3B parameters on 207B tokens–representing, to our knowledge, the largest implicit model trained to date. Notably, our implicit models outperform their explicit counterparts on standard benchmarks. Our code is publicly available at github.com/microsoft/implicit_languagemodels.

## 1. Introduction

Transformers, despite their dominance on contemporary language benchmarks, exhibit fundamental limitations in computational expressiveness. Both theoretically and empir-

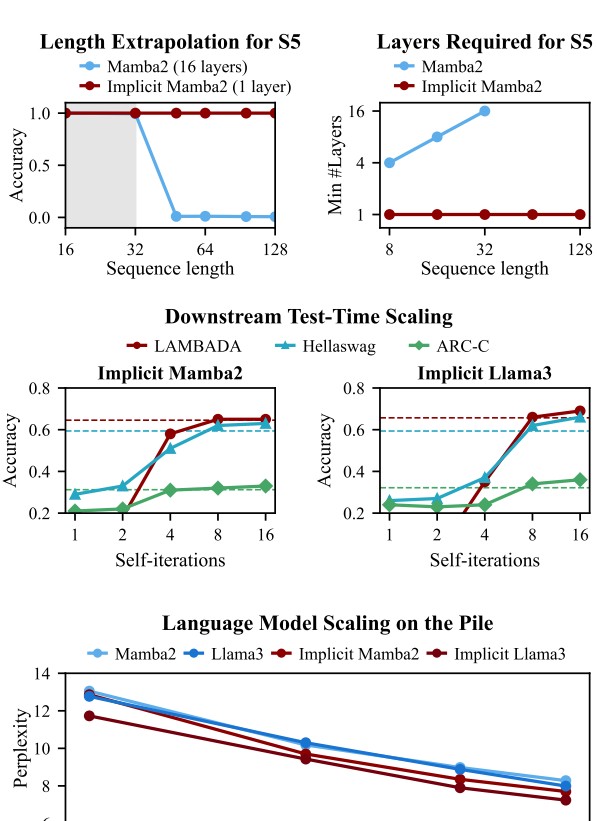

Figure 1: **Top Left:** Minimum layers required to solve the $S_5$ word problem, a theoretically hard formalization of state tracking, for different sequence lengths. **Top Right:** Length generalization for Mamba2 and our implicit Mamba2 trained on $L = 32$ and extrapolated up to $L = 128$. **Center:** Downstream task accuracy for a range of self-iterations at test-time for our implicit 1.3B parameter Mamba2 and Llama3 models. Dashed lines indicate explicit models' accuracy. **Bottom:** Scaling of language models pretrained on 207B tokens of the deduplicated PILE.

ically, they cannot fully recognize regular languages (Bhattamishra et al., 2020) or, equivalently, represent finite state machines (FSMs) (Merrill et al., 2022). This limitation is significant because FSMs form the backbone of many real-world state-tracking problems, including evaluating code,

---

[*]Equal contribution  [1]Chair of Highly-Parallel VLSI Systems and Neuro-Microelectronics , TUD Dresden University of Technology, Dresden, Germany [2]Microsoft Research, Cambridge, United Kingdom. Correspondence to: Jannes Gladrow <jannes.gladrow@microsoft.com>.

tracking object permutations (e.g., in games like chess or structured narratives), and modeling sequential dependencies in logic (Li et al., 2021), location tracking (Guan et al., 2023), games (Li et al., 2023) and scientific applications such as protein generation, genetics, and chemistry (Briand et al., 2023; Chowdhury et al., 2022; Boiko et al., 2023). This raises questions about the ability of transformers to maintain coherent world models based on transitions between states (Vafa et al., 2024) and hence, their suitability for tasks requiring robust state-tracking. These shortcomings appear to stem from a fundamental trade-off between parallelizability at training time and the ability to track state (Merrill & Sabharwal, 2023).

Surprisingly, recently emerging state-space models (SSM), a class of *linear* recurrent neural networks, are bound by the same trade-off: despite their seemingly sequential nature they cannot express some inherently sequential problems such as certain regular languages (Merrill et al., 2024). In contrast, non-linear recurrent neural networks (RNNs) are not bound by these restrictions on compute complexity and can track state (Siegelmann & Sontag, 1992; Merrill, 2019) but lack parallelizability at scale. This raises the question: *How much sequential processing does one have to accept to solve the state tracking problem?*

Previous attempts to address these limitations in transformers have leveraged non-linear transitions through self-iteration in the depth dimension (Dehghani et al., 2019; Banino et al., 2021). However, backpropagation through unrolled networks is computationally prohibitive at scale. Deep equilibrium (DEQ) models (Bai et al., 2019), in contrast, define a function *implicitly* via the fixed-points of a neural network; their output is the result of self-iteration until convergence. Training such networks requires backpropagation solely at the fixed point, eliminating the need to traverse the iterative path and thereby decoupling memory usage from the depth of iterations. Emerging hardware promising rapid computation of fixed-points of neural networks (Brunner et al., 2025) may tilt the hardware lottery (Hooker, 2020) in favor of such implicit models, making this an opportune moment to explore their potential.

Our approach to balancing state tracking and parallelization relies on two key observations. First, we demonstrate that implicit models naturally adapt their compute load to the difficulty of the learning problem (see Figure 3**Left**). At both training and test time, such models effectively interpolate between their parallelizable form, when all tokens in the sequence are resolvable, and RNNs, when there are no resolvable tokens. Further, we show theoretically that implicit models have indeed non-linear token-to-token transitions similar to RNNs. Second, based on the success of transformers on many practical language modeling problems, we hypothesize that natural language contains only a

sparse set of tokens that cannot be resolved by transformers (and SSMs). Such non-solvable transitions are critical for state tracking but remain intractable for the class of circuits representable by transformers and SSMs (Merrill et al., 2022; 2024). Exploiting these properties, we devise implicit models that combine the expressive power of RNNs with the parallelizability of transformers and SSMs (see Figure 2). In contrast to conventional transformers and SSMs, implicit models can track state, even out-of-distribution (see Figure 1**Right**). In contrast to RNNs, these models permit a much larger degree of parallelization as the depth of self-iteration is much smaller than the sequence length (see Figure 3**Mid**).

**Contributions.** **(a)** We propose implicit SSMs and show theoretically that they represent non-linear and non-diagonal state-to-state transitions similar to RNNs. **(b)** We confirm empirically that implicit SSMs can solve the $S_5$ word problem, which conventional SSMs and transformers fail to solve. **(c)** We show by constructing distributions with varying difficulty level over the word problem that implicit SSMs as well as transformers require much fewer non-parallelizable transitions to learn word problems than RNNs **(d)** We demonstrate scalability of implicit models through a carefully chosen training curriculum that bounds the number of iterations, training implicit SSM and transformers up to 1.3B parameters on 207B tokens of the deduplicated PILE (D-PILE) (Gao et al., 2020)— see Figure 1**Bottom**, the largest self-iterated model with dynamic halting condition to date, to the best of our knowledge. **(e)** We highlight a set of properties of our pretrained implicit language models such as favorable length generalization, and path-independent auto-regressive generation.

## 2. Background

### 2.1. State-Space Models

SSMs are linear recurrent models which produce an output $y_t \in \mathbb{R}^{d_{\text{out}}}$ given an input $x_t \in \mathbb{R}^{d_{\text{in}}}$ and a sequentially updated hidden state vector $h_t \in \mathbb{R}^n$ via the recurrence

$$h_t = \Lambda(x_t)h_{t-1} + u(x_t) \quad (1)$$
$$y_t = f(h_{t-1}, x_t), \quad (2)$$

where $u$ and $f$ are possibly non-linear learned functions. The learned matrix $\Lambda \in \mathbb{R}^{n \times n}$ is typically diagonal and can be constant (Gu et al., 2022; Smith et al., 2023) or an input-dependent matrix-valued function (Qin et al., 2023; Gu & Dao, 2023; Dao & Gu, 2024). A SSM combines a number of these blocks with non-linear feed-forward blocks. In contrast to non-linear RNNs, the linear state recurrence (1) allows for training parallelism along the sequence dimension, and avoids the quadratic scaling of self-attention.

## 2.2. Limitations of Transformers and SSMs

Efficient parallelization is one of the central features enabling transformers and SSMs to scale to large machine learning problems such as language modeling. Parallel circuits, however, face fundamental trade-offs regarding the class of problems that they can address. In particular, transformers and SSMs theoretically fail to recognize certain regular languages, or equivalently, to simulate FSMs (Merrill et al., 2022; 2024). Empirical studies have confirmed that neither of the models are capable of learning the algorithms constituting certain regular languages (Bhattamishra et al., 2020; Sarrof et al., 2024). By contrast, the sequential nature of RNNs allows them to express all regular languages (Merrill, 2019). A detailed discussion is given in Appendix B.1.

## 2.3. Deep Equilibrium Models

Most deep learning architectures *explicitly* parametrize a function $x \mapsto y$ with a neural network. Deep Equilibrium Models (DEQ), in contrast, define a function *implicitly* via the fixed-points of an input-conditional neural network, i.e.,

$$z^* = F_\theta(z^*, x), \tag{3}$$

where $z^*$ is identified with the prediction $y$. Naively differentiating a loss function $\mathcal{L}(z^*)$ with respect to the model parameters $\theta$ generally requires a costly differentiation through the employed fixed-point solver. Instead, to allow for gradient computations with a constant memory footprint, DEQs utilize the Implicit Function Theorem:

Let $G_\theta(z, x) = z - F_\theta(z, x)$. If the Jacobian $J_{G,z}$ of $G$ w.r.t. $z$ is non-singular in $z^*$, then there exists an open set $U$ around $(x, \theta)$ and a unique function $\Phi$ on $U$ such that $\Phi(x, \theta) = z^*$ and $G(\Phi(\tilde{x}, \tilde{\theta}), \tilde{x}, \tilde{\theta}) = 0$ for all $(\tilde{x}, \tilde{\theta}) \in U$. Furthermore, the derivative of $\Phi$ w.r.t. $\theta$ is given by

$$\frac{\partial \Phi}{\partial \theta} = -J_{G,z^*}^{-1} \frac{\partial F_\theta}{\partial \theta}. \tag{4}$$

A range of methods have been proposed to efficiently compute $\frac{\partial \mathcal{L}}{\partial \theta} = \frac{\partial \Phi}{\partial \theta} \frac{\partial \mathcal{L}}{\partial z^*}$ using Equation (4) (Bai et al., 2019; Geng et al., 2021). Here, we employ the Phantom Gradient approach of Geng et al. (2021) (see in the Appendix Figure 5). The method is based on solving a smoothed version of the fixed point Equation (3) combined with a finite truncation of the von Neumann series of the Jacobian-vector-product in (4) given as

$$\widehat{\frac{\partial \Phi}{\partial \theta}} = \lambda \frac{\partial F_\theta}{\partial \theta}\Big|_{z^*} \sum_{i=0}^{k-1} \left(\lambda \frac{\partial F_\theta}{\partial z}\Big|_{z^*} + (1-\lambda)I\right)^i, \tag{5}$$

where a small smoothing parameter $\lambda \in (0, 1]$ helps maintaining a small condition number at the cost of increased fixed-point iterations and the truncation length $k$ determines the accuracy of the approximation.

## 3. Implicit Sequence Models

### 3.1. Implicit State-space Models

The linear recurrence of SSMs shown in Equation (1) cannot resolve elaborate sequential problems (Merrill et al., 2024). Here, we propose to exploit self-iterations along the depth of neural networks to close the expressivity gap between SSMs and RNNs. Following the DEQ paradigm (see Section 2.3), we implicitly define a model via the fixed points of a SSM. Introducing the iteration variable $z_t^{(s)} \in \mathbb{R}^{d_{\text{out}}}$ to Equations (1) and (2) yields the fixed point iteration

$$h_t^{(s)} = \Lambda\left(z_t^{(s-1)}, x_t\right) h_{t-1}^{(s)} + u\left(z_t^{(s-1)}, x_t\right) \tag{6}$$

$$z_t^{(s)} = f_\theta\left(z_t^{(s-1)}, h_{t-1}^{(s)}, x_t\right), \tag{7}$$

where $z_t^{(0)} = 0$ for $t = 0, \ldots, T$ and $h_0^{(s)} = 0$ for $s = 0, \ldots, S$ respectively. In practice, we rely on the Mamba2 (Dao & Gu, 2024) architecture and inject the expanded input $x_t$ additively after the initial projection.

Computing the fixed-points and the gradient around the fixed points of Equations (6) and (7) requires to iterate the two loops $t = 1, \ldots, T$ along the input sequence and $s = 0, \ldots, S$ to converge to fixed points along depth. Figure 2 visualizes how these two loops give rise to two distinct modes of evaluation. The *simultaneous mode* simultaneously finds the fixed points for all $t$ (see Figure 2**A**), and exploits parallelization strategies for SSMs (Dao & Gu, 2024). The *sequential mode* resolves the $s$ and $t$ loops in the transpose order, and processes sequences sequentially just like classical SSMs or RNNs (see Figure 2**B**). While the simultaneous mode allows for highly parallel training, the sequential mode enables efficient inference at constant memory, e.g. for language generation. The emergence of these two strategies for finding the fixed points sparks the question if they are interchangeable as visualized in Figure 6. Can a model be trained in the simultaneous mode and deployed for inference in the sequential mode? This question is answered in Section 5 and Figure 2 for real language models.

For both modes, Equations (6) and (7) in the limit $s \to \infty$ read

$$h_t^* = \Lambda\left(z_t^*, x_t\right) h_{t-1}^* + u\left(z_t^*, x_t\right) \tag{8}$$

$$z_t^* = f_\theta\left(z_t^*, h_{t-1}^*, x_t\right), \tag{9}$$

where $z_t^* = \lim_{s \to \infty} z_t^{(s)}$ and $h_t^* = \lim_{s \to \infty} h_t^{(s)}$ denote the fixed points. Equations (8) and (9) couple the outputs $z_t^*$ with the states $h_t^*$ via the non-linear functions $f_\theta$, $\Lambda$ and $u$. Notably, what appears as a minor technical modification of the original state-space model in Equations (1) and (2) leads to a fundamentally enhanced expressivity of the recurrent model.

Figure 2: **A:** The simultaneous mode self-iterates the entire sequence such that trajectories interact during convergence. It exploits the parallelism of the backbone model. **B:** The sequential mode iterates each token individually. Only converged hidden states or kv-caches are passed on. This mode is used for generation. **C:** Difference in perplexity between the two modes for our 1.3B implicit models.

**Theorem 1.** *An implicit SSM defined by Equations* (8) *and* (9) *with generic weights yields a non-linear and non-diagonal state-to-state transition function* $h_{t-1}^* \mapsto h_t^*$. *If further* $z_t^* - f_\theta\left(z_t^*, h_{t-1}^*, x_t\right) = 0$ *is non-singular w.r.t.* $z_t^*$, *then there exists a continuously differentiable function* $\varphi$ *such that* $z_t^* = \varphi\left(h_t^*, x_t, \theta\right)$. *In this case, the state-to-state Jacobian is given by*

$$\left.\frac{dh_t^*}{dh_{t-1}^*}\right|_{z_t^*, x_t, \theta} = \Lambda\left(z_t^*, x_t\right) + \frac{\partial\Lambda}{\partial z_t^*}\frac{\partial\varphi}{\partial h_{t-1}^*}\mathrm{diag}\left(h_{t-1}^*\right)$$
$$+ \frac{\partial u}{\partial z_t^*}\frac{\partial\varphi}{\partial h_{t-1}^*}. \tag{10}$$

*Proof:* We refer the reader to Appendix A.2 for the proof and an extended statement of the theorem. A numerical check of Equation (10) is provided in Figure 7. □

As discussed in Section 2.2, non-linear RNNs surpass transformers and linear SSMs in terms of circuit complexity. By the above construction, our implicit SSM exhibits the favourable computational properties of RNNs, lifting the illusion of state in linear SSMs (Merrill et al., 2024). Furthermore, the gradients of a fixed point iteration depend solely on the fixed point, and not on the path to the fixed point, by the implicit function theorem. This suggests that both modes resolving the two for loops yield functionally equivalent fixed points.

These properties raise the following hypotheses, which we will investigate empirically in this work.

*Hypothesis* 1 (Expressivity). Implicit SSMs can learn and express all regular languages.

*Hypothesis* 2 (Parallelization). Implicit SSMs can be trained in simultaneous mode and evaluated in sequential mode without loss in performance.

### 3.2. Implicit Transformers

Similar to implicit SSMs, one can define an implicit transformer model (Bai et al., 2019) by injecting the input $x_t$ into

the attention operator as $\mathrm{Attn}(z_t W_{QKV} + x_t W_{\mathrm{inp}})$, where $W_{QKV} \in \mathbb{R}^{d\times 3d}$ produces the $Q$, $K$, $V$ for the multi-head self-attention (Attn), and $W_{\mathrm{inp}} \in \mathbb{R}^{d\times 3d}$ is the input projection. Conventional transformers, with their finite number of layers, cannot learn certain formal languages outside of the $\mathrm{TC}^0$ circuit complexity class (Merrill et al., 2022; Strobl et al., 2024). However, chain of thought (CoT) models (Wei et al., 2022) bypass this restriction by using an adaptive compute budget through recursive generation of intermediate tokens (Merrill & Sabharwal, 2024). Implicit transformers (Bai et al., 2019) utilize an adaptive compute budget differently, using fixed-point iterations that can be interpreted as sequences of latent thoughts (Hao et al., 2024), undergoing non-linear updates similar to a non-linear RNN's hidden state.

## 4. Implicit SSMs Adapt to Hard Languages

**Implicit SSMs Lift the Illusion of State**  The illusion of state (Merrill et al., 2024) reveals that constant depth SSMs cannot simulate arbitrary finite state machines. A hard state tracking problem in the sense that all state tracking problems can be reduced to it is given by the *word problem* for the symmetric group $S_5$ (Barrington, 1989). The word problem for a monoid $(M, \circ)$ is to resolve arbitrary length products of the form $\hat{m} = m_1 \cdot m_2 \circ \cdots \circ m_k$ for $m_1, m_2, \ldots, m_k \in M, k \in \mathbb{N}$. A comprehensive introduction to the word problem and our particular learning setting is provided in Appendix C.

We train a set of Mamba2 SSMs (Dao & Gu, 2024) to reproduce the results of Merrill et al. (2024). Figure 1**Left** highlights that Mamba2 requires more layers as the sequences get longer. For example resolving sequences of 32 elements from $S_5$ requires a minimum of 16 layers. Extending the result of Merrill et al. (2024), Figure 1**Right** shows that the same Mamba2 model with 16 layers does not generalize beyond the training distribution when evaluated on sequences longer than 32 elements. Our implicit Mamba2, however, can utilize additional self-iterations at test-time to

resolve longer sequences of up to 128 elements. This result establishes that implicit SSMs effectively learn to be RNNs. However, with naive unrolling in implicit SSMs, parallelization would still be challenging. In the following, we show a subtle yet important result: Implicit SSMs can adapt to word problems of varying difficulty even when trained with bounded depth.

**Languages with Sparse Non-Solvable Transitions** SSMs excel in natural language processing tasks despite being theoretically constrained to the simple class of star-free formal languages (Sarrof et al., 2024). We conject that natural language is mostly composed of simple to comprehend tokens, while harder tokens appear only sparsely. To study implicit models in a controlled learning environment closer to natural language than the $S_5$ word problem, we construct a word problem that mixes simple and hard examples. Let $M = M^a \times G$ be a direct product of an aperiodic monoid $M^a$ and a non-solvable group $G$. A sequence $m_0, \ldots, m_T$ is sampled from $M$ with replacement. To control the number of hard examples and simple examples, we define a family of distributions $\mathcal{D}_p$ over $M$ as follows. An element $m_k^a \in M^a$ is sampled uniformly at each step $k$, representing the presence of simple examples. On the other hand, we sample elements $g_k \in G \setminus \{e\}$ from $G$ without the identify transformation, each with probability $\frac{p}{|G|-1}$. The identity element $g_k = e \in G$ is sampled with probability $1 - p$. The resulting transformations $(m_k^a, g_k)$ are aperiodic at least when $g_k = e$, i.e. with probability $1 - p$.

**Interpolating between SSMs and RNNs** We will identify minimally sequential models that parallelize to a high degree and still capture all non-solvable transitions in a language. Therefore, we apply our construction of a word problem above to mix tokens from simple languages with tokens from non-solvable hard languages. This section studies a word problem over $M = M^a \times A5$, where $M^a$ is a simple aperiodic monoid with four elements and $A_5 \subset S_5$ is the alternating group over 5 elements, the smallest non-solvable subgroup of $S_5$. For details on the learning problem, we refer the reader to Appendix C.

We train Mamba2 and implicit Mamba2 models on a range of mixtures of simple and hard tokens between $p = 0.0$ and $p = 0.25$, and in the case of the implicit models with varying self-iteration depths at training time between 2 and 128. All training sequences sample $L = 256$ tokens, and evaluation is conducted on the distribution $\mathcal{D}_{0.5}$, where half of the tokens is hard. The evaluation is hence an out-of-distribution (OOD) setting. We report averaged results over 10 random seeds with boostrapped $95\%$ confidence intervals as well as the best models per configuration. None of the conventional models got OOD accuracies beyond random chance as shown in the right panel of Figure 3, hence we will

focus our discussion on the implicit models in the following. The left panel of Figure 3 shows that implicit SSMs capture the underlying algorithm, as measured by out-of-distribution evaluation with $p = 0.5$, even when trained on very few non-solvable tokens. While a fraction of $2\%$ hard tokens per sample ($p = 0.02$) suffices for some configurations, reliable training can be observed from $p = 0.1$ on.

We are left with the question of how many self-iterations are required during training to learn the algorithm intrinsic to the word problem. To answer this we trained a range of models with $p = 0.1$, setting a different upper bound on the number of self-iterations at training time. The number of self-iterations at test time is unbounded and solely defined by the fixed point iteration. The mid panel of Figure 3 shows that a small amount of down to 8 self-iterations at training time suffices to generalize from the distribution $\mathcal{D}_{0.1}$ at training time to $\mathcal{D}_{0.5}$ at test time. Interestingly, the number of test time self-iterations is quite similar for the models trained with different upper bounds on the training time self-iterations, hinting that the models learned similar algorithms. Note that the self-iterations required during training are significantly lower than the sequence length. For comparison, a conventional RNN conducts $L = 256$ non-parallelizable steps to solve the same problem, a factor of 32 larger than the 8 self-iterations required by our implicit Mamba2. This comes at a cost: we need to self-iterate over every token. However, each self-iteration can be parallelized across the sequence dimension by the parallelization of the base model.

In the right panel of Figure 3, we demonstrate that the phantom gradient is, in most cases, a more effective method for gradient computation than backpropagation through the entire sequence of unrolling steps. To evaluate this, we train three variants of the Mamba2 model: (1) an implicit Mamba2, which self-iterates and employs phantom gradients; (2) an unrolled Mamba2, which backpropagates through all unrolling steps; and (3) an explicit Mamba2, a conventional model. All models are trained on sequences of length $L = 256$ sampled from $\mathcal{D}_{0.1}$, with a depth constraint of 16 – corresponding to 16 self-iterations for the implicit and unrolled models and 16 layers for the explicit model. Our result shows that a constant number of backpropagation steps using the phantom gradient method is enough to learn complex non-solvable transitions and generalize to difficult distributions at test time. Since phantom gradients require a constant memory that is independent of the number of self-iteration steps, the training of larger language models appears feasible.

**CatbAbi: A benchmark requiring state tracking.** To evaluate the state-tracking capabilities of SSMs on language tasks, we use the CATBABI dataset (Schlag et al., 2021), a modified version of the BABI dataset (Weston et al., 2015),

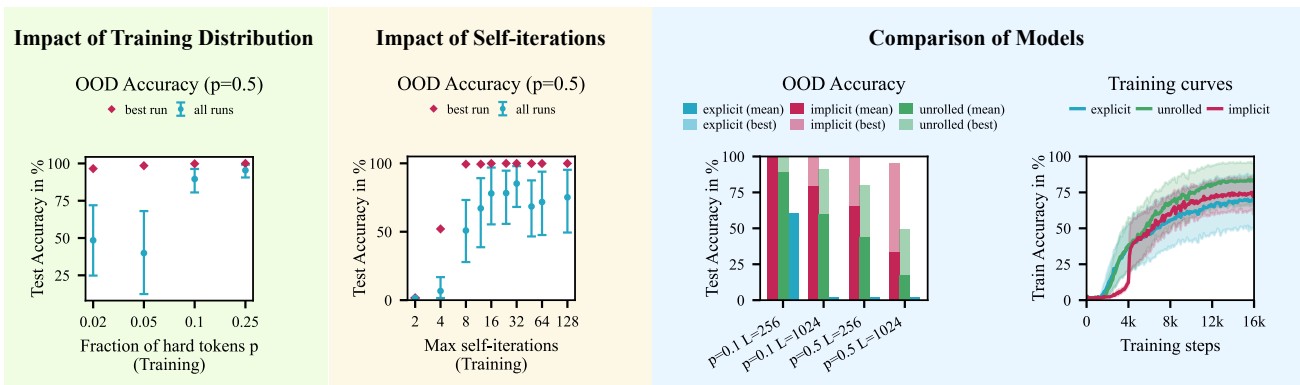

Figure 3: All models were trained and evaluated on sequences of length $L = 256$. The out-of-distribution (OOD) evaluation is conducted with $p = 50\%$. **Left:** Comparison of OOD accuracy for a range of training distributions with hard token probabilities $p$. **Mid:** Comparison of OOD accuracy for a range of self-iterations caps at training time, trained with $p = 0.1$. **Right:** Comparison of implicit Mamba2, unrolled Mamba2, and Mamba2 trained with $p = 0.1$. Unrolled Mamba2 unrolls a single layer with full backpropagation, while implicit Mamba2 receives only 4 Phantom Gradient steps. All models have a training depth of 16 (layers for Mamba2, self-iterations for implicit and unrolled).

consisting of 20 tasks within a 5M token corpus. These tasks, requiring various reasoning abilities like deduction, conference, or counting, involve short stories with embedded questions (Schlag et al., 2021), and require state tracking in various degrees. We train our implicit SSM model, using Mamba2 as the core architecture, alongside the baseline Mamba2 model, both with up to three layers. Our findings show that the implicit Mamba2 model with a single layer outperforms its single-layer Mamba2 counterpart on most tasks. Additionally, more layers in the implicit model's backbone reduce the number of self-iteration steps needed to solve the tasks (see Appendix Figure 13a, Figure 13b). We furthermore evaluate the performance of the models for tasks sorted by increasing story length. We see how implicit models retain its performance as the lengths increases in Figure 13c at a slight increase in the number of iterations of the implicit models in Figure 13d.

## 5. Implicit Large Language Models

We investigate whether implicit models can be effectively pretrained to function as language models. Motivated by the results of Section 4, we implement a pretraining strategy for implicit models with two stages of bounded and free self-iterations. Transformer (LLama) Touvron et al. (2023) and SSM (Mamba2) (Dao & Gu, 2024) architectures serve as the core backbones for our implicit models. In the bounded stage, we train with four self-iterations and a single step of phantom gradient, which we refer to as the $(4 + 1)$-model. The $(s + k)$-notation refers to $s$ gradient-tape-free self-iteration steps and $k$ phantom gradient steps. $k$ refers to Equation (5), see also Figure 5. The free stage starts from a checkpoint of the $(4 + 1)$-model and increases the number

of self-iterations to 24/32 followed by four steps of phantom gradient. We refer to these models as $(24 + 4)/(32 + 4)$-models for Mamba2/Llama, respectively. We employ four model sizes: 125M, 350M, 760M, and 1.3B. These models are pretrained in an autoregressive manner for next-token prediction across all sizes on the D-PILE (Gao et al., 2020) dataset, which consists of 207B tokens. For baselines, we use both Mamba2 (Dao & Gu, 2024) and Llama (Beck et al., 2024) models previously trained on a corpus of 300B tokens. Additionally, we reproduce Mamba2* and Llama† as baselines trained with the same code and data as our implicit models. We evaluate the pretrained models on the test set of the D-PILE, examine their length extrapolation capabilities, and assess their common sense reasoning performance on downstream tasks. See Appendix D.3 for pretraining details.

**Pretraining Results and Downstream Performance.** We report in Table 1 the next-token perplexity performance of all models trained on the entire 207B token corpus using a test split of the D-PILE[1]. We observe our implicit models consistently achieve a lower perplexity compared to their explicit counterparts—see also Figure 1**Bottom**. For details related to the dynamics of the implicit models on D-PILE, refer to Table 6. Additionally, we evaluate the models' performance on common sense reasoning tasks using the LM Evaluation Harness (Gao et al., 2024). The results show that implicit Mamba2 outperform the explicit Mamba2*, which are pretrained on the same number of tokens, on most tasks. This difference becomes more pronounced as the size of the models increases, specifically with the 760M

---

[1]The test split represents a random selection of 0.1 percent of the entire dataset. This size is in line with the proportion used for the PILE's validation set (Gao et al., 2020).

and 1.3B variants. Compared to the original Mamba2 baseline, trained on 1.5 times more data, the implicit models do better on HELLASWAG, PIQA, ARC-E, and are competitive in LAMBADA and ARC-C. Across all scales, the implicit Mamba2 models significantly outperform Mamba2 in the HELLASWAG task, yet they underperform in WINO-GRANDE and OPENBOOKQA.

It is also noteworthy that our implicit Llama models substantially outperform the baseline Llamas, including both the results reported in (Beck et al., 2024) and the Llama†. This improvement is consistent across all tasks and model sizes. Strikingly, we note that our implicit Llama (32+4) 760M is competitive to the explicit Llama† 1.3B.

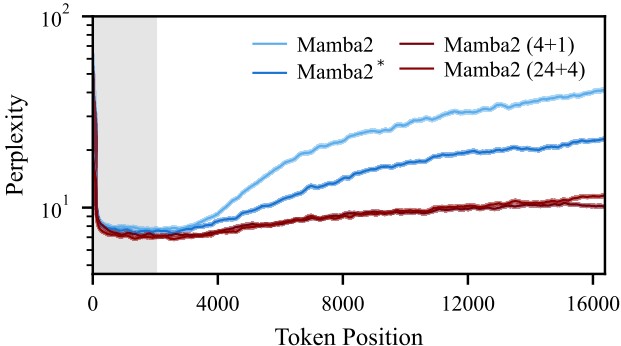

Figure 4: Length extrapolation performance of the per token perplexity on the the test split of the D-PILE of the original 1.3B Mamba2, our Mamba2*, and our implicit Mamba2 with (4+1) and (24+4) self-iterations. Shaded gray area shows the in-distribution length.

**Implicit-SSMs Demonstrating Length Extrapolation Capabilities**   All implicit models in our study were trained on sequences of 2048 tokens. To assess their capability for length extrapolation, we evaluated the implicit models on the test split of the D-PILE, which was packed with longer sequences consisting of 4096, 8192, and 16384 tokens. We compared these results with the baseline Mamba2 and Mamba2* in Figure 4, where the per-token perplexities are reported. Numerical values for all models at a range of token positions are reported in Table 7 in the Appendix. In comparison with the baseline Mamba2 models, our implicit Mamba2 models demonstrate stronger robustness on longer sequences.

**Effective Duality between Simultaneous Mode and Sequential Mode**   Autoregressive generation, a core functionality of contemporary language models, for implicit models requires that the sequential mode introduced in Section 3 and Figure 2 is functionally equivalent to the simultaneous mode used for pretraining. Effectively, the loops over $s$ and $t$ in Equation (6) have to be interchangeable (also

see Figure 6), which we empirically demonstrate with our pretrained language models. Specifically, we utilize our 1.3B implicit Mamba2 (24+4) and Llama (32+4) models to compute next-token predictions on the D-PILE test split. The models are fed identical input tokens of length 2048 in batches of size 16 and predict outputs greedily in both simultaneous and sequential modes. We observe token match rates of 97.6 % (on 3M tokens) between the outputs of the two modes for the implicit Mamba2, and 97.7 % (on 330K tokens) for the implicit Llama. Examples of these model predictions are provided in Appendix Table 8. The per-token perplexity differences in the predictions of the models are depicted in Figure 2. To our knowledge, this is the first demonstration of sequential evaluation with self-iterated models at constant memory in the number of self-iterations, enabling auto-regressive generation for this class of models.

## 6. Related Work

**Adaptive-Compute Time**   The idea of an adaptive compute budget goes back to (Schmidhuber, 2012) who employ a halting neuron to delimit the computation on a particular input. Graves (2017) generalized the idea and regularised the halting condition to encourage the network to stop early. They implemented an adaptive-depth RNN and demonstrated the network adjusting the compute budget based on the difficulty of instances in a parity-check task. This idea was later applied to Transformers, resulting in "Universal Transformers" (UT) (Dehghani et al., 2019). UTs can either be unrolled to a fixed depth or augmented with a dynamic halting condition (DHC) per token. UTs were later shown to exhibit improved scaling laws compared to standard transformers (Kaplan et al., 2020). PonderNet (Banino et al., 2021) introduced a principled probabilistic model for determining the halting condition. This approach improved on the UT on the BABI benchmark. Recently, a mixture-of-experts (MoE) variant of the UT (MoEUT) was presented (Csordás et al., 2024) with 1B parameters, seeking to improve the parameter-to-compute ratio of UTs. The MoEUT is an unrolled model with fixed iterations and does not employ a DHC. While our models presented here are dense, they could, in principle, be turned into MoE. Gatmiry et al. (2024) show that looped linear transformers implement gradient-descent until convergence on the prediction loss defined by previous input-output examples in the context window. Lim et al. (2024) take the opposite approach to our work: Instead of augmenting SSMs or transformers, they propose an approach based on fixed-point iterations to enable parallel training of RNNs. However, their method incurs cubic cost in terms of state size, limiting the method to smaller models.

**Reasoning and out-of-distribution generalization.**   The ability of looped models to generalize better to input lengths

Table 1: Comparison of test set perplexity and downstream performance. We compare our implicit models, which have 4 self-iteration steps and 1 phantom gradient step (denoted as 4+1), and those with 24/32 self-iteration steps and 4 phantom gradient steps (denoted as 24+4/32+4), with our baseline models Mamba2* and Llama†. These baseline models as well as the implicit models are trained on 207B tokens from the D-PILE dataset and range in size from 130M to 1.3B parameters. For further comparison, we include the original Mamba2 (Dao & Gu, 2024) (trained on 300B tokens of the PILE) and the Llama (trained on 300B tokens of the SLIMPAJAMA) from (Beck et al., 2024). The best performing model for each type is highlighted in bold, and the second-best is underlined. The training wall-clock time (WCT) is provided for implicit models relative to their explicit counterpart. For $(24 + 4)$- and $(32 + 4)$-iterated models, we provided the weighted average measured over the entire curriculum. We omit WCTs for the 370M-model series since the runs were carried out on various node sizes, making direct comparison difficult.

| | Model | Dataset/ Tokens (B) | Rel. WCT Impl/Expl | D-Pile ppl↓ | LAMBADA ppl↓ | LAMBADA acc↑ | HellaSwag acc↑ | PIQA acc↑ | Arc-E acc↑ | Arc-C acc↑ | WinoGrande acc↑ | OpenbookQA acc↑ | Average acc↑ |
|---|---|---|---|---|---|---|---|---|---|---|---|---|---|
| 130M | Mamba2 | Pile/300 | - | 13.72 | 16.83 | 0.4388 | 0.3525 | 0.6496 | 0.4739 | 0.2423 | 0.5233 | 0.306 | 0.4266 |
| | Mamba2* | D-Pile/207 | 1 | 13.05 | 18.51 | 0.4116 | 0.3527 | 0.6572 | 0.4815 | 0.2372 | 0.5130 | 0.300 | 0.4219 |
| | Mamba2(4+1)-ours | D-Pile/207 | 1.97 | 13.76 | 18.58 | 0.4118 | 0.3628 | 0.6485 | 0.4537 | 0.2287 | 0.5107 | 0.288 | 0.4149 |
| | Mamba2(24+4)-ours | D-Pile/207 | 3.27 | 12.86 | 18.03 | 0.4174 | 0.3673 | 0.6496 | 0.4604 | 0.2372 | 0.5178 | 0.290 | 0.4200 |
| | Llama | SlimPajama/300 | - | - | 39.21 | 0.3154 | 0.3409 | 0.6545 | 0.4533 | 0.2363 | 0.5067 | - | 0.4178 |
| | Llama† | D-Pile/207 | 1 | 12.77 | 17.08 | 0.4297 | 0.3513 | 0.6540 | 0.4794 | 0.2440 | 0.5122 | 0.280 | 0.4215 |
| | Llama (4+1)-ours | D-Pile/207 | 0.90 | 12.73 | 15.54 | 0.4518 | 0.3706 | 0.6447 | 0.4823 | 0.2372 | 0.5391 | 0.290 | 0.4308 |
| | Llama (32+4)-ours | D-Pile/207 | 2.08 | 11.73 | 13.39 | 0.4801 | 0.3958 | 0.6676 | 0.4886 | 0.2355 | 0.5304 | 0.298 | 0.4423 |
| 370M | Mamba2 | Pile/300 | - | 10.55 | 8.00 | 0.5593 | 0.4692 | 0.7046 | 0.5476 | 0.2671 | 0.5564 | 0.324 | 0.4897 |
| | Mamba2* | D-Pile/207 | - | 10.18 | 8.96 | 0.5333 | 0.4653 | 0.6942 | 0.5526 | 0.2696 | 0.5320 | 0.306 | 0.4790 |
| | Mamba2(4+1)-ours | D-Pile/207 | - | 10.02 | 8.79 | 0.5457 | 0.4684 | 0.6899 | 0.5358 | 0.2696 | 0.5162 | 0.308 | 0.4762 |
| | Mamba2(24+4)-ours | D-Pile/207 | - | 9.70 | 8.26 | 0.5575 | 0.4792 | 0.7040 | 0.5484 | 0.2688 | 0.5351 | 0.316 | 0.487 |
| | Llama | SlimPajama/300 | - | - | 15.73 | 0.4419 | 0.4445 | 0.6915 | 0.5223 | 0.2628 | 0.5359 | - | 0.4832 |
| | Llama† | D-Pile/207 | - | 10.30 | 8.37 | 0.5624 | 0.4537 | 0.6844 | 0.5476 | 0.2577 | 0.5541 | 0.318 | 0.4826 |
| | Llama (4+1)-ours | D-Pile/207 | - | 9.66 | 7.03 | 0.5898 | 0.5030 | 0.7024 | 0.5539 | 0.2611 | 0.5572 | 0.314 | 0.4973 |
| | Llama (32+4)-ours | D-Pile/207 | - | 9.43 | 7.04 | 0.5956 | 0.5114 | 0.7078 | 0.5244 | 0.2705 | 0.5722 | 0.320 | 0.5003 |
| 760M | Mamba2 | Pile/300 | - | 9.23 | 5.86 | 0.6167 | 0.5492 | 0.7198 | 0.6103 | 0.2850 | 0.6030 | 0.362 | 0.5351 |
| | Mamba2* | D-Pile/207 | 1 | 8.98 | 6.24 | 0.6125 | 0.5418 | 0.7231 | 0.6044 | 0.2858 | 0.5777 | 0.338 | 0.5262 |
| | Mamba2(4+1)-ours | D-Pile/207 | 2.05 | 8.60 | 6.15 | 0.6117 | 0.5569 | 0.7296 | 0.6077 | 0.3140 | 0.5509 | 0.336 | 0.5295 |
| | Mamba2(24+4)-ours | D-Pile/207 | 3.43 | 8.35 | 5.90 | 0.6191 | 0.5698 | 0.7334 | 0.6090 | 0.3131 | 0.5730 | 0.338 | 0.5365 |
| | Llama | SlimPajama/300 | - | - | 9.90 | 0.5141 | 0.5216 | 0.7095 | 0.5648 | 0.2875 | 0.5667 | - | 0.5274 |
| | Llama† | D-Pile/207 | 1 | 8.88 | 5.77 | 0.6375 | 0.5448 | 0.7171 | 0.5905 | 0.2816 | 0.6054 | 0.338 | 0.5307 |
| | Llama (4+1)-ours | D-Pile/207 | 1.64 | 8.27 | 5.15 | 0.6524 | 0.5853 | 0.7312 | 0.6052 | 0.3097 | 0.5967 | 0.356 | 0.5481 |
| | Llama (32+4)-ours | D-Pile/207 | 2.97 | 7.90 | 4.82 | 0.6703 | 0.5995 | 0.7416 | 0.6187 | 0.3012 | 0.5991 | 0.344 | 0.5535 |
| 1.3B | Mamba2 | Pile/300 | - | 8.40 | 5.02 | 0.6559 | 0.5995 | 0.7378 | 0.6418 | 0.3319 | 0.6117 | 0.378 | 0.5652 |
| | Mamba2* | D-Pile/207 | 1 | 8.28 | 5.12 | 0.6456 | 0.5939 | 0.7416 | 0.6145 | 0.3123 | 0.6117 | 0.352 | 0.5531 |
| | Mamba2(4+1)-ours | D-Pile/207 | 1.91 | 7.97 | 5.21 | 0.6383 | 0.6136 | 0.7437 | 0.6343 | 0.3302 | 0.5746 | 0.354 | 0.5555 |
| | Mamba2(24+4)-ours | D-Pile/207 | 3.19 | 7.70 | 4.99 | 0.6489 | 0.6267 | 0.7416 | 0.6423 | 0.3336 | 0.5888 | 0.352 | 0.5620 |
| | Llama | SlimPajama/300 | - | - | 7.23 | 0.5744 | 0.5781 | 0.7312 | 0.6279 | 0.3174 | 0.5904 | - | 0.5699 |
| | Llama† | D-Pile/207 | 1 | 7.99 | 4.95 | 0.6569 | 0.5936 | 0.7432 | 0.6385 | 0.3217 | 0.6062 | 0.352 | 0.5589 |
| | Llama (4+1)-ours | D-Pile/207 | 1.96 | 7.66 | 4.40 | 0.6852 | 0.6397 | 0.7448 | 0.6338 | 0.3396 | 0.6575 | 0.360 | 0.5801 |
| | Llama (32+4)-ours | D-Pile/207 | 3.91 | 7.24 | 4.24 | 0.6901 | 0.6583 | 0.7465 | 0.6654 | 0.3601 | 0.6401 | 0.364 | 0.5892 |

not seen during training is empirically well established: For example Yang et al. (2024a) show this for looped transformers, while Anil et al. (2022) demonstrate length generalization for DEQs, particularly when they are path independent. Du et al. (2022) show that energy-based models trained to map energy-gradient-descent steps to algorithmic steps, can length generalize in summation, and complex algorithms such as shortest-path. On the theoretical side, The pioneering work of Siegelmann & Sontag (1992) shows that iterated RNNs are Turing complete at infinite numerical precision. More recently, Deletang et al. (2023) studied a number of sequence models and report that grouping tasks by their rung in the Chomsky hierarchy is predictive of models ability to length-generalize. While the works of Merrill *et al* (Merrill, 2019; Merrill et al., 2020; Merrill & Sabharwal, 2023; Merrill et al., 2024), which we discuss inSection 2.2, showed that both transformers and SSMs are restricted to $TC^0$; several studies sought to find more precise constraints. Weiss et al. (2021) observe that programs written in a specific language (RASP) can be mapped to transformer models of sufficient capacity. Zhou et al. (2023) then showed that transformers tend to length-generalise if the underlying data-generating process can be expressed in RASP. Sarrof et al. (2024) derived a similar refined constraint for SSMs and showed that they can precisely express star-free regular languages. Grazzi et al. (2025) demonstrate that SSMs can track state in simple problems, such as parity, when their (diagonal) recurrence matrix $\Lambda$ in Equation (1) permits negative eigenvalues. Moreover, they illustrate that a variant of DeltaNet (Yang et al., 2024b) with (possibly) negative eigenvalues can solve the S5 problem when only swaps of two values are considered in the transition. However, no variant of Mamba or DeltaNet was capable of learning S5 and achieving length generalization. To tackle the parallelization-expressiveness trade-off, Beck et al. (2024) propose two new LSTM-inspired layer archi-

tectures: the sLSTM and mLSTM layers. While the latter is parallelizable, the former is not and intended to enable the whole model to recognize regular languages. Finally, Soulos et al. (2024) survey strategies for chunking input sequences with transformers, maintaining parallelizability within each chunk and using RNN-like transitions between chunks. They find these architectures recognize regular languages for small chunk sizes with scaling remaining a challenge.

## 7. Discussion and Conclusion

This work demonstrates that models implicitly defined by a fixed point iteration can solve hard state tracking problems that resist the capabilities of transformers and SSMs. We provide theoretical insight how implicit SSMs can deviate from pure diagonal and linear token-to-token transitions and effectively become an RNN in the limit. When trained with a relatively small number of self-iterations, our models seamlessly generalize from simpler to harder word problems (see Figure 3). This property is of special interest in language modeling where 'hard' sequences are rare but might occur clustered in applications requiring state tracking.

Our extensive study of synthetic state tracking problems informs a pretraining schedule for large language models. The implicit Llama and Mamba2 models improve over the baselines in many cases, and prove particularly beneficial on downstream tasks such as HELLASWAG (see Table 1). Performance on language modeling is typically primarily determined by parameter count which traditionally caused weight-shared models to underperform (Tay et al., 2023). While implicit models lift the state-tracking limitations of explicit language models, state-of-the-art explicit models perform extraordinarily well on natural language. Self-iteration introduces additional cost that might only amortize on specific problems requiring state-tracking over long sequences. However, emerging hardware that accelerates such self-iteration would alleviate this overhead (Brunner et al., 2025). Furthermore, as LLMs make more progress on reducing perplexity, they may eventually face tokens requiring RNN-like transitions.

Finally, given the recent rise of test-time compute (Snell et al., 2025) and latent-space reasoning (Hao et al., 2024), models with adaptive depth per token deserve careful consideration as potential bridgeheads for such techniques as they natively offer adaptive depth and latent-space iteration.

## Impact Statement

This paper presents work aimed at advancing the field of machine learning by developing sequential and language models that can track state and hence prove to be more powerful in reasoning. Regarding language modeling, there are numerous potential societal consequences to consider, such as the energy consumption required for the training of large models and the additional iterations that our implicit models impose (see Table 4) to enhance computational expressivity of language models. When developing training algorithms for our language modeling, computational efficiency was taken into account— see Appendix D.3. Nevertheless, further research is essential to maintain the power of the proposed models while continuing to reduce computational costs.

## Acknowledgment

The authors of the paper would like to thank colleagues from the Analog Optical Computer (AOC) team at Microsoft Research Cambridge for their discussions and feedback during the project. Additionally, we acknowledge support from the Microsoft GCR team for providing the GPUs and prompt assistance in resolving issues faced during the training of large language models. MS was partially supported with funds from Bosch-Forschungsstiftung im Stifterverband.

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

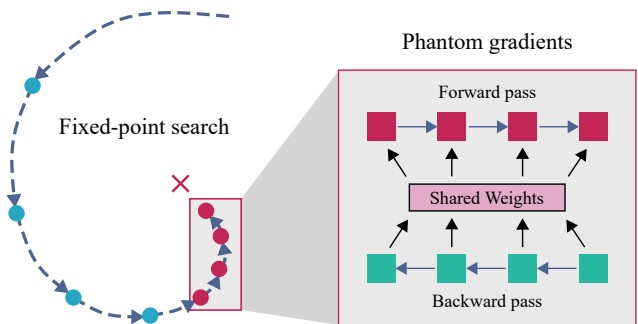

Figure 5: Fixed-point iteration and phantom gradients: A neural network is iterated until convergence in the forward pass. When employing the phantom gradient principle, only the final $k$ steps of the forward pass are considered for the backward pass as expressed in Equation (5). Therefore, the memory required for the backward pass is proportional to $k$ and in particular not proportional to the number of steps in the forward pass.

## Duality between Simultaneous and Sequential Modes

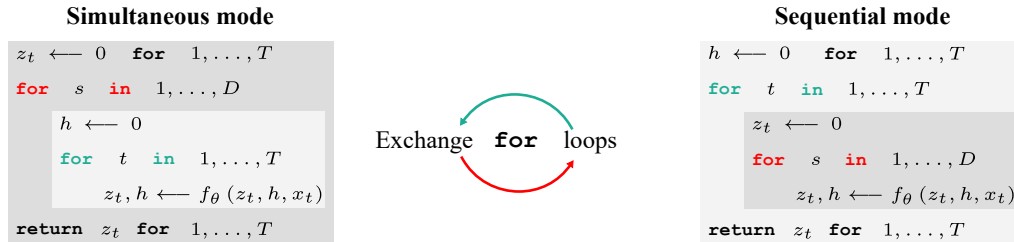

Figure 6: The simultaneous mode exploits the parallelism of state-space models or transformers, while the sequential mode is well suited for language generation. State-space models can further utilize the sequential mode for processing with constant memory over any sequence length. The two modes emerge from exchanging the for loops over the two variables $t$ and $s$ in the DEQ iteration (6). We demonstrate in Section 5, and Figure 2, that 1.3B parameter language models trained with the simultaneous mode show negligible difference in perplexity when evaluated with the sequential mode.

## A. Implicit State-Space Models

### A.1. Fixed Point Iteration

Figure 5 visualizes the forward and backward pass of implicit models. A fixed point search is conducted by unrolling the model until the relative difference $\frac{|z_t^{(s)} - z_t^{(s-1)}|}{|z_t^{(s-1)}|}$ falls below a tolerance $\epsilon$, where we choose $\epsilon = 0.05$ for the language models and $\epsilon = 0.01$ for the state-tracking models.

The phantom gradient approach backpropagates only through a fixed number of steps, effectively decoupling the number of iterations in the forward pass from the number of iterations in the backward pass.

Figure 6 shows the two loops over the sequence $t$ and over depth $s$ as presented in Section 3.1. The equations involving the iteration variables are printed again below.

$$h_t^{(s)} = \Lambda\left(z_t^{(s-1)}, x_t\right) h_{t-1}^{(s)} + u\left(z_t^{(s-1)}, x_t\right)$$
$$z_t^{(s)} = f_\theta\left(z_t^{(s-1)}, h_{t-1}^{(s)}, x_t\right).$$

## A.2. Proof of Theorem 1

**Theorem 1** (Extended). *An implicit SSM defined by Equations* (8) *and* (9) *with generic weights yields a non-linear and non-diagonal state-to-state transition function* $\xi : h_{t-1}^* \mapsto h_t^*$. *Let* $g(z, h, x, \theta) = z - f_\theta(z, h, x)$. *If* $g\left(z_t^*, h_{t-1}^*, x_t, \theta\right) = 0$ *and* $J_{g,z}|_{h_{t-1}^*, x_t, \theta} = \left.\frac{\partial g}{\partial z}\right|_{h_{t-1}^*, x_t, \theta}$ *is non-singular, there exists an open set* $U$ *with* $\left(h_{t-1}^*, x_t, \theta\right) \in U$ *and a continuously differentiable function* $\varphi$ *on* $U$ *s.t.*

$$g\left(\varphi(h, x, \theta), h, x, \theta\right) = 0 \quad \forall (h, x, \theta) \in U .$$

*In this case, the state-to-state Jacobian* $J_\xi$ *is given by*

$$\left.\frac{dh_t^*}{dh_{t-1}^*}\right|_{z_t^*, x_t, \theta} = \Lambda\left(z_t^*, x_t\right) + \frac{\partial \Lambda}{\partial z_t^*}\frac{\partial \varphi}{\partial h_{t-1}^*}\mathrm{diag}\left(h_{t-1}^*\right) + \frac{\partial u}{\partial z_t^*}\frac{\partial \varphi}{\partial h_{t-1}^*} .$$

*Proof.* Suppose that fixed points $h_{t-1}^*, z_{t-1}^*$ have been found for $t-1$. Our goal is to back the above statements for $t$. Let $T(z) = f_\theta\left(z, h_{t-1}^*, x_t\right)$. With generic weights $\theta$, e.g. $\theta$ in general linear position, a multi-layer neural network $f_\theta$ is non-linear and its input-output function $T(z)$ has a non-diagonal Jacobian. Finding the fixed point $z_t^*$ can be expressed in terms of recursive application of the non-linear operator $T$ as

$$z_t^* = \lim_{s \to \infty} z_t^{(s)} = \lim_{s \to \infty} T^s\left(z_t^{(0)}\right) , \tag{11}$$

where we set $z_t^{(0)} = 0$. By extension from multi-layer networks to self-iterated multi-layer networks, generic weights yield a non-linear function $\left(h_{t-1}^*, x_t, \theta\right) \longrightarrow z_t^*$. Once the fixed point $z_t^*$ is found, we can compute $h_{t-1}^*$ by a single pass through Equation (8).

To proof Equation (10), we apply the implicit function theorem to the function

$$g(z, h, x, \theta) = z - f_\theta(z, h, x) . \tag{12}$$

If $g\left(z_t^*, h_{t-1}^*, x_t, \theta\right) = 0$ and $J_{g,z}|_{h_{t-1}^*, x_t, \theta} = \left.\frac{\partial g}{\partial z}\right|_{h_{t-1}^*, x_t, \theta}$ is non-singular, the implicit function theorem guarantees the existence of an open set $U$ with $\left(h_{t-1}^*, x, \theta\right) \in U$ and a continuously differentiable function $\varphi$ on $U$ s.t.

$$g\left(\varphi(h, x, \theta), h, x, \theta\right) = 0 \quad \forall (h, x, \theta) \in U . \tag{13}$$

The derivative of $\varphi$ at the fixed point is given by

$$\left.\frac{\partial \varphi}{\partial h}\right|_{h_{t-1}^*, x_t, \theta} = \left(I - \left.\frac{\partial f_\theta}{\partial z}\right|_{z_t^*, h_{t-1}^*, x_t, \theta}\right)^{-1} \left.\frac{\partial f_\theta}{\partial h}\right|_{z_t^*, h_{t-1}^*, x_t, \theta} \tag{14}$$

By the above argument, $\varphi$ is a non-linear function if $f_\theta$ is a non-linear function, and $\varphi\left(h_{t-1}^*, x_t, \theta\right) = z_t^* = \lim_{s \to \infty} T^s(0)$.

Now, consider Equation (8) at the fixed point

$$h_t^* = \Lambda\left(z_t^*, x_t\right) h_{t-1}^* + u\left(z_t^*, x_t\right) , \tag{15}$$

where $z_t^* = \varphi\left(h_{t-1}^*, x_t, \theta\right)$. If $\varphi$ is a non-linear function of $h_{t-1}^*$, then $h_{t-1}^* \mapsto h_t^*$ is a non-linear function as well. Equipped with the derivative of $\varphi$, we can derive the state-to-state Jacobian of the implicit SSM as

$$\begin{aligned}
\left.\frac{dh_t^*}{dh_{t-1}^*}\right|_{z_t^*, x_t, \theta} &= \Lambda\left(z_t^*, x_t\right)\frac{\partial h_{t-1}^*}{\partial h_{t-1}^*} + \frac{\partial \Lambda}{\partial h_{t-1}^*}h_{t-1}^* + \frac{\partial u}{\partial h_{t-1}^*} \\
&= \Lambda\left(z_t^*, x_t\right) + \frac{\partial \Lambda}{\partial z_t^*}\frac{\partial \varphi}{\partial h_{t-1}^*}\mathrm{diag}\left(h_{t-1}^*\right) + \frac{\partial u}{\partial z_t^*}\frac{\partial \varphi}{\partial h_{t-1}^*} ,
\end{aligned}$$

where we used the implicit dependency of $z_t^*$ on $h_{t-1}^*$ via $\varphi$ to differentiate $\Lambda$ and $u$. $\square$

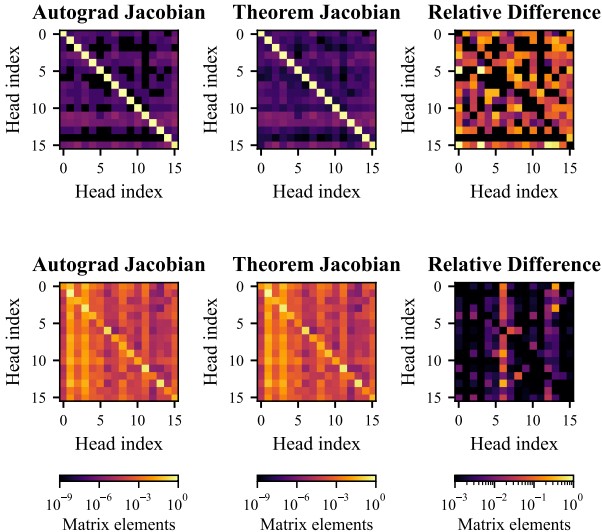

Figure 7: State-to-state Jacobian according to Equation (10) of a single layer implicit Mamba2. To simplify the visualization, only one state variable of each head is plotted. **Left:** Jacobian computed with automatic differentiation through the fixed point iteration. **Center:** Jacobian computed with Equation (10). **Right:** Relative difference $\frac{|J_{\text{autograd}} - J_{\text{theorem}}|}{|J_{\text{autograd}}| + |J_{\text{theorem}}| + \epsilon}$. **Top:** Randomly initialized model. **Bottom:** Trained model.

This equation (Equation (10) in the main text) highlights the non-diagonal corrections that the self-iteration of $z_t^{(s)}$ introduces to the diagonal Jacobian $\Lambda$ of the explicit state-space Equation (1). We conduct a numerical check of Equation (10) in Figure 7 on data from the synthetic task devised in Section 4. Therefore, the state $h_7^*$ is created by processing a sequence of 8 tokens $x_0, \ldots, x_7$. Next, we loop over $s$ until $z_8^{(s)}, h_8^{(s)}$ converge to a fixed point, where the stopping criterium is a relative difference $\frac{|z_8^{(s)} - z_8^{(s-1)}|}{|z_8^{(s-1)}|}$ of $1 \times 10^{-4}$. Figure 7 compares the Jacobian of $h_7^* \longrightarrow h_8^*$ when backpropagating through the iteration over $s$ with PyTorch's automatic differentiation (left column) versus Equation (10) (central column). The right column shows the relative difference between the two methods as $\frac{|J_{\text{autograd}} - J_{\text{theorem}}|}{|J_{\text{autograd}}| + |J_{\text{theorem}}| + \epsilon}$ with $\epsilon = 10^{-16}$. The top row shows a randomly initialized model, and the bottom row shows a trained model. Both models are based on the Mamba2 architecture with a single layer and $d_{\text{model}} = 64, d_{\text{head}} = 8, d_{\text{state}} = 4$ such that the model has 16 heads. A single state variable is selected per head, resulting in a $16 \times 16$ grid per model and evaluation method.

## B. Algebraic Structure of Finite State Machines

This section provides a basic introduction to the word problem and its relation to simulating finite state machines (FSMs). We start with some results in the circuit complexity and then relate them to the properties of FSMs.

### B.1. Circuit Complexity

Efficient parallelization is one of the central features enabling transformers and SSMs to scale to large machine learning problems such as language modeling. Parallel circuits, however, face fundamental trade-offs regarding the class of problems that they can address. Circuit complexity theory provides a framework to characterize the types of problems that parallel circuits can solve. $\text{TC}^0$ is the class of circuits with constant depth and polynomial width composed of unbounded fan-in AND-gates, OR-gates, NOT-gates and MAJORITY-gates. The second class of interest, $\text{NC}^1$, is represented by logarithmic depth circuits with a polynomial number of bounded fan-in gates. From the perspective of formal languages, $\text{NC}^1$ is equivalent to the class of circuits recognizing the regular languages. Since the unbounded fan-in gates allowed in $\text{TC}^0$ circuits can be constructed from log-depth circuits with bounded fan-in, it follows that $\text{TC}^0 \subset \text{NC}^1$. It is open if $\text{TC}^0$ is a proper subset of $\text{NC}^1$, and we will discuss a regular language for which no $\text{TC}^0$ circuit construction is known.

Both transformers and SSMs can be simulated by $TC^0$ circuit families under mild assumptions (Merrill et al., 2022; 2024). If $TC^0$ is a proper subset of $NC^1$, the leading sequence models today cannot even recognize all regular languages. Consequentially, they cannot execute arbitrary finite state machines (FSMs), a fundamental skill to execute tasks, or to form world models (Vafa et al., 2024). Many empirical studies confirm these theoretical limitations of transformers and SSMs to learn regular languages (Deletang et al., 2023; Sarrof et al., 2024; Strobl et al., 2024). At the same time, recurrent neural networks are known to recognize regular languages (Kleene, 1951; Elman, 1991; Merrill et al., 2020), and to effectively implement internal FSMs to solve language problems (Omlin & Giles, 1996).

### B.2. Algebraic Concepts of Finite State Machines

**Monoids and Groups**   There is a tight relationship between finite state machines and algebraic concepts such as monoid and groups. We define the relevant concepts for our state tracking problem described in Section 4

**Definition 2** (Monoid). A set $M$ and a binary operation $\circ : M \times M \longrightarrow M$ are called a *monoid* $(M, \circ)$ if

1. there exists an identity element $e \in M$ with $e \circ m = m \circ e = m$ for all $m \in M$

2. the operation $\circ$ is associative, i.e. $(m_1 \circ m_2) \circ m_3 = m_1 \circ (m_2 \circ m_3)$ for all $m_1, m_2, m_3 \in M$.

Straight forward examples for monoids are natural, rational or real numbers with multiplication, or strings with string concatenation. Since monoid are associative, we can simplify notation and write $m \circ m = m^2$, and so on for all powers $k \in \mathbb{N}$.

**Definition 3** (Aperiodic Monoid). A monoid $(M, \circ)$ is called *aperiodic* if for all $m \in M$ there is a $k \in \mathbb{N}$ s.t. $m^k = m^{k+1}$.

Monoid where each element has a a corresponding inverse element are called *groups*.

**Definition 4** (Group). A *group* $(G, \circ)$ is a monoid with the additional property that for every $g \in G$ there is $g^{-1} \in G$ s.t. $g \circ g^{-1} = g^{-1} \circ g = e$.

Examples for groups are rational numbers with multiplication, or the orthogonal matrices with matrix multiplication. Notably, permutations on a set of $k$ elements for $k \in \mathbb{N}$ form a group, called the *symmetric group* $S_k$.

Our synthetic learning problem discussed in Section 4 will be constructed based on a classical problem in computer science.

**Definition 5** (Word Problem). Let $M^*$ denote the set of all sequences over elements of $M$. The *word problem* on a monoid $(M, \circ)$ is defined by the function

$$\text{WP} : M^* \longrightarrow M$$
$$\text{WP}(m_0, m_1 \ldots, m_k) \mapsto m_0 \circ m_1 \circ \cdots \circ m_k, \tag{16}$$

i.e. a word over $M$ is resolved by composition to a single element in $M$.

The central question for our experiment will be which kinds of circuits can solve the word problem for arbitrary sequence lengths.

**Theorem 6** ((Barrington, 1989)). *The word problem for any fixed non-solvable group $G$ is complete for $NC^1$ under $AC^0$ reductions.*

**Algebra of FSMs**   Tracking the state of a system can be formalized as executing a finite state machine (Merrill et al., 2024). To characterize the limits of certain FSMs, we define a few formal concepts.

**Definition 7** (FSM). A *finite state machine* (FSM) consists of a finite set of states $Q$, a finite set of input symbols $\Sigma$ called the alphabet, and a transition function $\delta : Q \times \Sigma \longrightarrow Q$.

Given an initial state $q_0 \in Q$ and a sequence of symbols $w = a_1 a_2 \ldots a_k \in \Sigma^*$, a FSM transitions from the initial state into a final state.

Finite state machines naturally define a monoid.

**Definition 8** (Syntactic Monoid). For each symbol $a \in \Sigma$, define the function $\delta_a : Q \longrightarrow Q$. The transformation monoid $M$ generated by $\delta_a, a \in \Sigma$ and the composition of functions $\circ$, is called the *syntactic monoid* $(M, \circ)$ of the finite state machine.

| | | | |
|---|---|---|---|
| 1 | Yesterday Mary travelled to the kitchen. | 1 | Yesterday Bill went to the school. |
| 2 | Fred went to the park yesterday. | 2 | Mary went to the cinema yesterday. |
| 3 | This morning Fred journeyed to the bedroom. | 3 | This morning Julie went back to the kitchen. |
| 4 | Bill went back to the bedroom yesterday. | 4 | Yesterday Julie went to the office. |
| 5 | *Q*: Where was Fred before the bedroom? | 5 | *Q*: Where was Julie before the kitchen? |

Table 2: Two examples of tracking entities in natural language from the CATBABI dataset (Schlag et al., 2021).

The algebraic structure of $M$ is tightly coupled to the programs that the original FSM can execute. Our investigation is based on the classical result stated in Theorem 6. The simplest example of a non-solvable group is the permutation group of five elements $S_5$. A corollary from theorem 6 is that the FSM whose syntactic monoid is $S_5$ is complete in $NC^1$ and hence in the class of regular languages. We have thus identified a *hard* state tracking problem: Permutations of five elements.

Another classical result tightly related to state-space models is

**Theorem 9** ((Schützenberger, 1965))**.** *Let $L$ be the regular language defined by a FSM, and let $M$ be the syntactic monoid of the same FSM. Then $L$ is a star-free language if an only if $M$ is aperiodic.*

It is intuitive that the word problem for finite aperiodic monoids is in $TC^0$. The maximal depth of the circuit is driven by the number of elements of the monoid and it's maximal $k$ for the aperiodicity condition. Empirical studies have shown that transformers and SSMs can simulate a range of regular languages (Deletang et al., 2023; Strobl et al., 2024), but they struggle to learn the $S_5$ word problem in line with their characterization as $TC^0$ circuits (Merrill et al., 2024). SSMs can be further restricted to the star-free languages (Sarrof et al., 2024), i.e. those with aperiodic syntactic monoid.

## C. Relaxed Non-Solvable Word Problems

The $A5$ or $S5$ word problems used in (Merrill et al., 2024) require to keep track of state for every single token. However, entities appear sparsely in natural language as shown in Table 2. Consequently, tracking state of every token is not an accurate representation of the state tracking problem in natural language. Here, we devise a synthetic task to model tracking state of sparse entities. Let $p$ the sparsity of entities. For example $p = 0.9$ refers to 90 % sparsity, i.e. an entity density of 10 %. An example of a sparse sequence is presented in Figure 8. In particular, we are interested in answering the following questions

1. Which density of entities is required to learn the intrinsic algorithm of the state tracking task?

2. How many self-iterations do our implicit models need to conduct to learn the intrinsic algorithm?

To construct a learning problem for sequence models, we represent each element of the monoid $M$ as a token, and present a sequence $m_0, \ldots, m_L$ of tokens to the model. The ground truth at each position $k = 1, \ldots, L$ is the token representing the element $m_0 \circ \cdots \circ m_k$. We then calculate the mean cross entropy loss over the entire sequence, providing a learning signal at each step $k = 1, \ldots, L$.

State-space models can learn the word problem for aperiodic monoids (Sarrof et al., 2024), but fail so solve it for non-solvable groups such as $S_5$ (Merrill et al., 2024). We confirm in Figure 1 that implicit state-space models can in fact learn the word problem for $S_5$. We now want to *interpolate between word problems for aperiodic and non-solvable monoids* to test how much signal our implicit state-space model defined in Section 3 needs from the hard non-solvable group word problem to learn it.

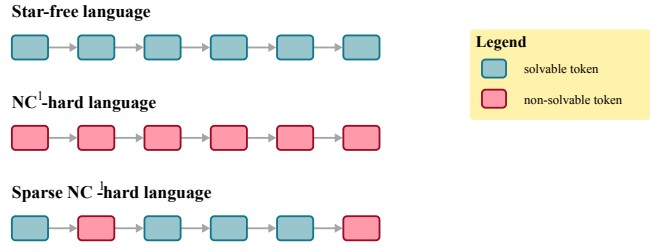

Figure 8: Examples of word problems with mixed solvable and non-solvable tokens. We refer to tokens from a non-solvable group as non-solvable tokens, and to tokens from a star free language as solvable tokens.

Let $M = M^a \times G$ be a direct product of an aperiodic monoid $M^a$ and a non-solvable group $G$. A sequence $m_0, \ldots, m_T$ is sampled from $M$ with replacement. To control the number of hard examples and simple examples, we define a family of

## Model Comparison

Figure 9: Comparison of implicit Mamba2, unrolled Mamba2, and Mamba2 for $p = 0.05$. All models were trained and evaluated on sequences of length $L = 256$. Unrolled Mamba2 refers to a single layer being unrolled multiple times with a full backpropagation trace, while the implicit Mamba2 receives only 4 steps of Phantom Gradient. The training time depth of all models is limited to 16, i.e. 16 layers for Mamba2, and 16 self-iterations for the implicit and weight tied models. Implicit and unrolled models use unbounded test-time computation to converge to a fixed point. The comparison shows that the implicit model with 4 steps of Phantom Gradient succeeds over the unrolled model.

distributions $\mathcal{D}_p$ over $M$ as follows. An element $m_k^a \in M^a$ is sampled uniformly at each step $k$, representing the presence of simple examples. On the other hand, we sample elements $g_k \in G \setminus \{e\}$ from $G$ without the identify transformation, each with probability $\frac{p}{|G|-1}$. The identity element $g_k = e \in G$ is sampled with probability $1 - p$. The resulting transformations $(m_k^a, g_k)$ are aperiodic at least when $g_k = e$, i.e. with probability $1 - p$.

We'll call the tokens representing $(m, e)$ *simple tokens* and the tokens representing $(m, g)$ with $g \neq e$ are called *hard tokens*. The names derive from the fact that SSMs can resolve the word problem if it is only composed from simple tokens. Any non-zero probability $p$ of sampling hard tokens renders the word problem unsolvable for fixed depth SSMs on arbitrarily long sequences.

Our construction of a distribution over a monoid allows us to test out-of-distribution generalization not only in terms of length generalization, the most common setting in the literature. By changing $p$ between training time and test time, we construct training tasks and evaluation tasks with varying difficulty. This effectively offers OOD evaluation with the same number of tokens, but different mixtures of easy and hard tokens. While this property allows us to distil expressivity questions from length generalization properties, our construction is not limited to the same sequence length and could as well be used in the length generalization setup (see Figure 9).

## D. Experimental Details

### D.1. The Word Problem

Each data point in Figure 3 and Figure 9 is based on 10 runs with different random seeds. We report the best run, mean accuracy and a 95 % confidence interval for each data point. All word problem models were trained on sequences of length $L = 256$, and a batch size of 512 on 32 GB V100s. The explicit models and self-iterated models with full backpropagation trace required gradient accumulation over two steps. The learning rate is set to 0.001. We disable dropout and weight decay, which appears to harm learning on the word problem. The self-iterations are stopped upon convergence, which we define as a relative difference between two consecutive states of 0.01 for Figure 3 or 0.05 for Figure 9. We trained a number of standard Mamba2 models with the same number of runs for multiple numbers of layers. These models struggle to capture the training distribution compared to self-iterated models, and none of them was able to generalize to a harder distribution or to longer sequences.

### D.2. CatbAbI

The models, both implicit and explicit, comprise up to three layers of the Mamba2 architecture with an embedding dimension of 256, a state dimension of 16 (expansion factor 2), and a head dimension of 32. We trained the models using batch sizes of 128 and 256, and learning rates of 0.0001, 0.0005, 0.001, and 0.005. The models were trained for 15,000 steps, with the implicit model specifically trained for 5,000 steps in unrolling mode, utilizing 32 steps with normal gradient checkpointing, followed by 10,000 steps of self-iteration fixed-point search. The self-iteration included a stop threshold of 0.03 and a training and testing maximum number of steps 50 and 200 , respectively, and phantom gradient parameters of 6 steps with ($\lambda = 0.5$). Data were packed in sequences of length 200 tokens as per (Schlag et al., 2021). Figures 10 and 11 illustrate the validation accuracy of the explicit and implicit Mamba2 models on the CATBABI dataset, respectively. Additionally, Figure 12 plots the number of iterations required for the implicit model to reach a fixed point on the validation set of the CATBABI dataset.

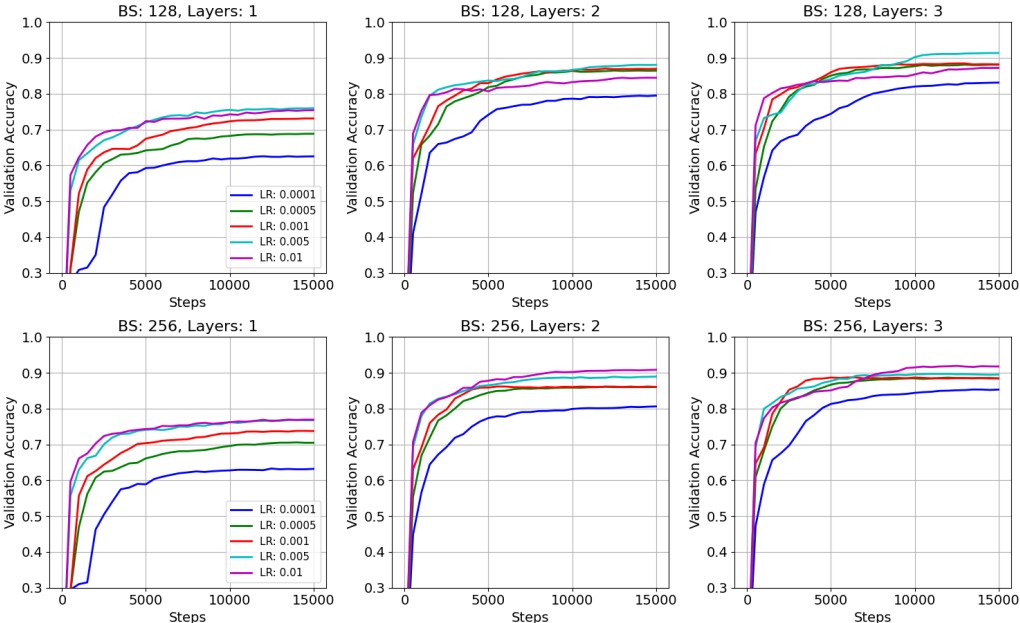

Figure 10: Hyperparameter sweeps for explicit Mamba2 models over batch sizes 128, 256, layers 1,2,3 and various learning rates for training on the CATBABI dataset.

### D.3. Language Modeling

**Pretraining Details**   We have trained a suite of Implicit SSM models with the core architecture of Mamba2 and Implicit Transformer models with the core of Llama3. For each implicit model, we have a corresponding weight-tied model that is also trained on the entire D-PILE dataset. We use the checkpoint from 80 percent of the way through training the weight-tied model to train the fully implicit model. We use four scales for the training of the models: 1.3B, 760M, 360M, and 130M. In all models, the LLM head weights are tied to the embedding weights. The implicit and weight-tied models have the same architecture as those of the Mamba2 and Llama models, except for an injection module, consisting of an MLP, which transforms the input into the domain of the mixer latent space. This module has a constant size equivalent to $2 \times d_{emb} + 2 \times d_{state} + n_{heads}$ in Mamba2, matching with $d_{in_{proj}}$, and $3 \times d_{emb}$ in Llama models, corresponding to the key, value, and queries, and is shared across all layers of the model. Details for each model is provided in Table 3.

We followed the training recipe of Mamba2 and Llama models. In particular, we used a weight decay of 0.1, no bias for the LLM head, AdamW hyperparameters $\beta = (0.9, 0.95)$, RMSNorm instead of LayerNorm, and a linear warm-up step to the peak learning value, which is chosen as 5 times the value of the GPT-3 model. For the learning rate scheduler, we used a constant learning rate followed by a square root decay to a minimum value of $10^{-5}$ (Hägele et al., 2024). While this scheduling has also been shown to be compute-optimal (Hägele et al., 2024) alongside the cosine scheduling, it allows us to use intermediate checkpoints during training more conveniently without considering how the new learning rate affects

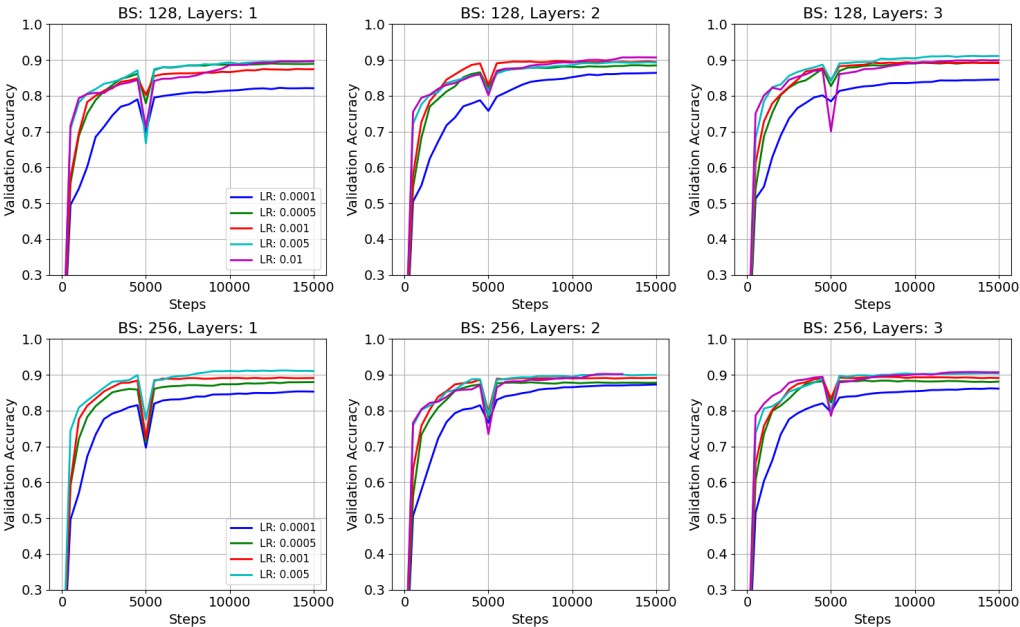

Figure 11: Hyperparameter sweeps for implicit Mamba2 models over batch sizes 128, 256, layers 1,2,3 and various learning rates for training on the CATBABI dataset.

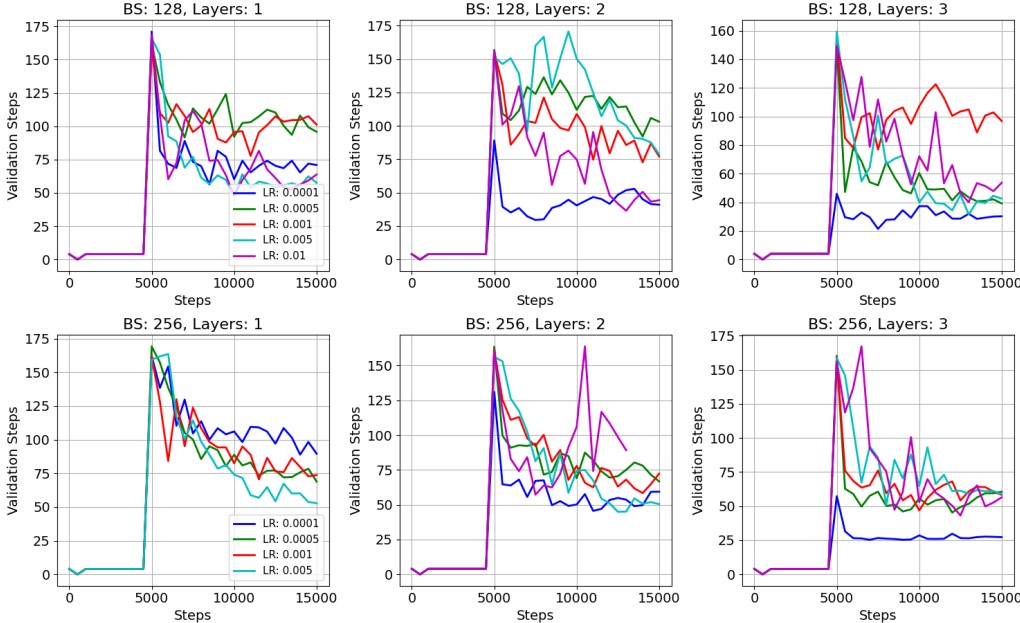

Figure 12: Hyperparameter sweeps for implicit Mamba2 models over batch sizes 128, 256, layers 1,2,3 and various learning rates for training on the CATBABI dataset.

the training of implicit models. All models were trained with an effective batch size of 1M tokens and a training sequence length of 2048 tokens.

**Downstream Task Details** We evaluated our models as well as the baseline models on seven tasks: LAMBADA_OPENAI (Paperno et al., 2016), HELLASWAG (Zellers et al., 2019), PIQA (Bisk et al., 2020), ARC-EASY and ARC-CHALLENGE (Clark et al., 2018), WINOGRANDE (Sakaguchi et al., 2021), and OPENBOOKQA (Mihaylov et al., 2018). We used the

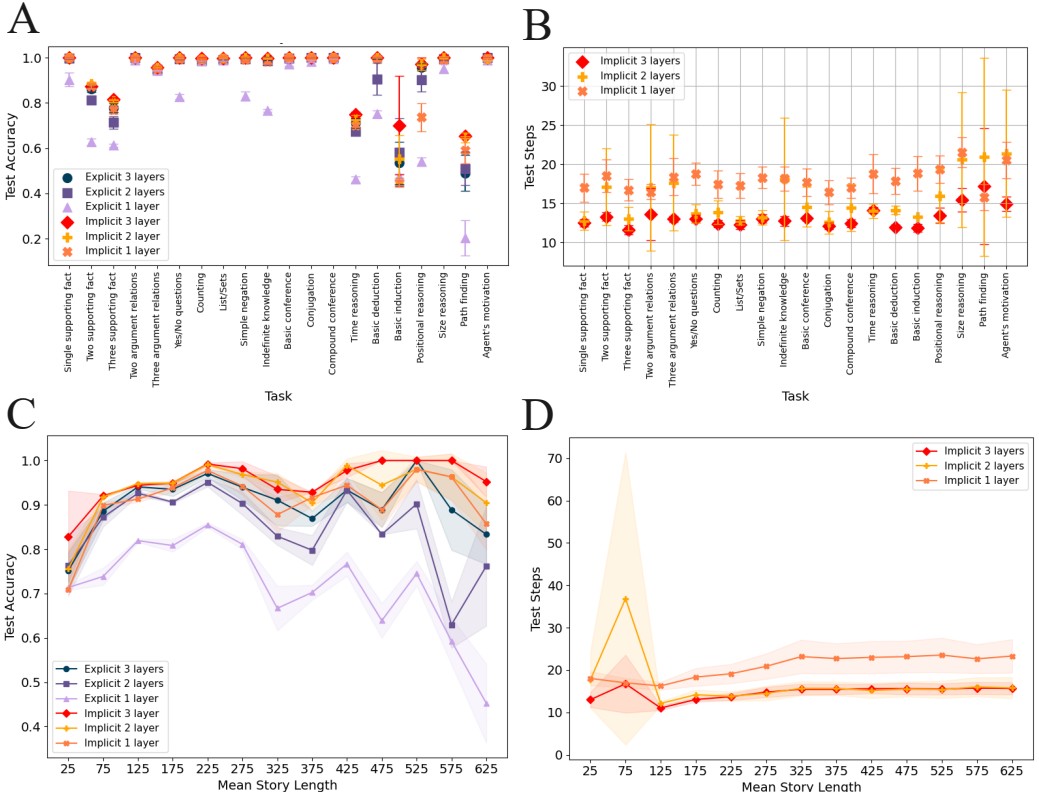

Figure 13: Small-scale reasoning advantages of implicit SSMs compared to explicit SSMs on the CATBABI dataset. (**a**) Task-specific performance comparison, measured in accuracy of the models in predicting answers to questions within a story, between the implicit Mamba2 and baseline Mamba2. The one-layer implicit Mamba2 model outperforms the one-layer explicit Mamba2 on most tasks. As the number of layers in the explicit Mamba2 increases, its performance approaches that of the implicit Mamba2. Adding more layers to the implicit Mamba2 benefits certain tasks, such as 'Basic induction' or 'Path finding,' where the implicit Mamba2 achieves the best performance. (**b**) The correlation between the number of self-iteration steps that the implicit Mamba2 takes to solve a task and the number of layers in the implicit model's backbone architecture, showing a decrease in the required steps with additional layers. (**c**) The implicit Mamba2 retains its performance as the story length increases, whereas the explicit Mamba2's performance declines. (**d**) The trend in the number of iterations needed by the implicit Mamba2 models as story length increases, indicating a modest rise in computational steps.

model checkpoints on the 207B tokens of the D-PILE. For the evaluation of the downstream tasks, we utilized the LM Evaluation Harness package (Gao et al., 2024) with the default specifications, i.e., a batch size of 256 and a sequence length of 2048. All models were evaluated using one H100 80GB GPU.

**Curriculum-Based Training of Language Models** The training of implicit models is achieved through a curriculum training framework that is divided into two main phases: bounded phase and the free phase. In the bounded phase, the models are subjected to four steps of self-iteration, followed by a singular phantom gradient step to store the activations necessary for backpropagation. This phase of training involves 80 percent of the dataset (the choice of this proportion and its influence on model performance are discussed below). Training then progresses to the free phase, wherein the model undergoes additional fixed point searches, capped at 24 self-iterations for SSMs and 32 for Transformers, and is followed by four phantom gradient iterations. A stopping criterion of $\epsilon = 0.05$ is implemented during this second phase, allowing models to terminate the fixed point search once this threshold is met. For validation and testing on the D-PILE dataset, the limit on fixed-point iterations is set to four times the number used in the free phase of training. The learning rate reduction

Table 3: Architecture and pretraining details

| Params. | SSM/Transformer # layers | Dim. Emb. | # Heads/ Dim. | Training Steps | Peak Learning Rate | Batch Size (Tokens) |
|---|---|---|---|---|---|---|
| 1.3B | 48/24 | 2048 | 32/64 | 197701 | 0.001 | 1M |
| 760M | 48/24 | 1536 | 16/96 | 197701 | 0.00125 | 1M |
| 350M | 48/24 | 1024 | 16/64 | 197701 | 0.0015 | 1M |
| 130M | 24/12 | 768 | 12/64 | 197701 | 0.003 | 1M |

starts in phase two; for Transformers, this reduction begins immediately to prevent instability caused by their high spectral norm. Conversely, for SSMs, the learning rate cooldown starts after 90 percent of the overall training period has elapsed, due to their spectral norm being approximately one, which permits more substantial weight adjustments at a higher learning rate.

**Duration of Bounded and Free Phases in Training Language Models**   In our exploration of the optimal duration for bounded and free phases of training, we aimed to find a balance between computational efficiency and the necessary nonlinear transitions each token must undergo through self-iteration. We tested models trained with 70, 80, and 90 percent of the bounded phase duration before starting the full fixed-point search in the free phase—refer to Fig. 14a. For this evaluation, we used a 130M Mamba2 model. Based on the model's perplexity on the validation split of D-PILE (2M examples of length 2048), we observed that beyond a certain extent of free phase training, the model's performance plateaus or overfits. We determined that 20 to 10 percent of free phase training is optimal. Consequently, we applied 20 percent free phase training for the training of larger models. It is crucial to note that the 130M model, when trained with an effective batch size of 0.5M tokens, experienced overfitting. This overfitting was not present when we increased the batch size to 1M tokens, —see Fig. 14b. Thus, we adopted a 1M token batch size for the training of models across all scales.

**Specification of Fixed Point Solver for Language Model Training**   Our experimentation with the fixed point solver in the free phase involved adjusting various parameters, including the maximum number of self-iterations for the fixed point search (16, 24, 32) and the number of gradient accumulation steps (2, 4). The 32 iterations have a smaller stop threshold of 0.02, whereas the 16 and 24 iterations have a stop threshold of 0.05——see Fig. 14b. We used a 130M Mamba2 model for this evaluation with an effective batch size of 1M tokens. We measured the validation split perplexity during training. For the training of the implicit Transformer models, we used a maximum self-iteration cap of 32. This was due to the higher spectral norm of the Transformer models requiring more steps to reach a fixed point below the threshold.

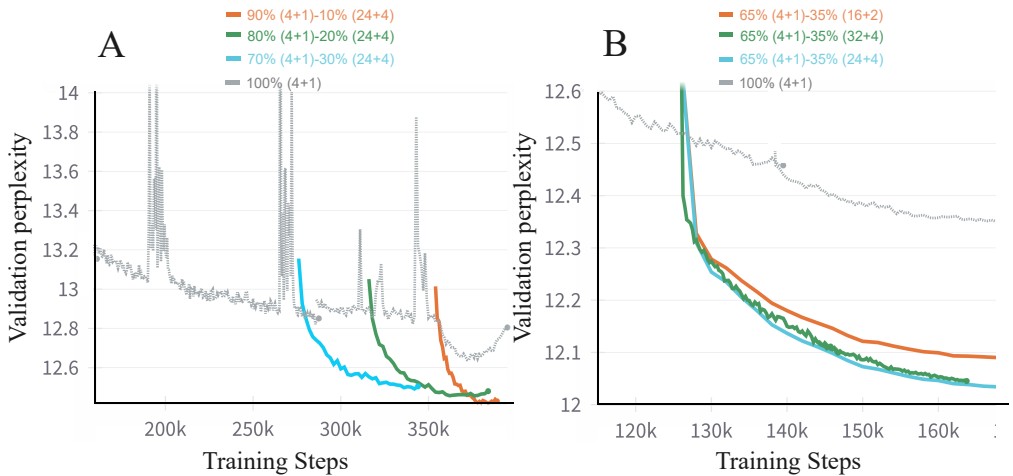

Figure 14: **Left**: Impact of Bounded Phase (4+1) Training Duration on Model Performance: A comparison of perplexity on the validation split of the D-PILE obtained by 130M Mamba2 models trained with 70%, 80%, and 90% bounded phase durations, with a batch size of 0.5M tokens. **Right**: Evaluation of Fixed Point Solver Specifications: The relationship between different maximum iterations (16, 24, 32) and gradient accumulation steps (2, 4) on the validation split perplexity of the D-PILE for the 130M Mamba2 model with a batch size of 1M tokens.

**Resource Allocation for Training and Evaluating Large Language Models**    We trained our suite of models on a cluster with AMD Instinct MI300X GPUs. Each node within the cluster comprises 8 GPUs, and we employed distributed multi-node processing to train our models on up to 32 GPUs simultaneously. Table 4 details the number of GPUs allocated for the training of each model, as well as the total GPU hours consumed by each. The evalution of models on downstream tasks was achieved on one 80GB Nvidia H100 GPU.

Table 4: GPU resource allocation and utilization for training large language models. The GPU counts employed and the total GPU hours expended for the training of each Mamba2 and Llama model variant across different parameter scales—130M, 350M, 760M, and 1.3B is listed. The models were trained using a cluster of AMD Instinct MI300X GPUs, with 8 GPUs per node, utilizing distributed multi-node processing with up to 32 GPUs in parallel.

| | Model | GPU Counts | Total GPU Hours |
|---|---|---|---|
| **130M** | Mamba2* | 8 | 1620 |
| | Mamba2 (4+1) | 8 | 3185.6 |
| | Mamba2 (24+4) | 8 | 2756.8 |
| | Llama† | 32 | 1146.56 |
| | Llama (4 + 1) | 32 | 1027.2 |
| | Llama (32+4) | 32 | 1561.6 |
| **350M** | Mamba2* | 32 | 1516 |
| | Mamba2 (4+1) | 8 | 2427.2 |
| | Mamba2 (24+4) | 8 | 2168.8 |
| | Llama† | 32 | 1324.8 |
| | Llama (4 + 1) | 8 | 2515.2 |
| | Llama (32+4) | 8 | 2335.2 |
| **760M** | Mamba2* | 16 | 1920 |
| | Mamba2 (4+1) | 16 | 3932.8 |
| | Mamba2 (24+4) | 16 | 3440 |
| | Llama† | 16 | 4636.8 |
| | Llama (4 + 1) | 16 | 7612.8 |
| | Llama (32+4) | 32 | 7676.8 |
| **1.3B** | Mamba2* | 32 | 3054.4 |
| | Mamba2 (4+1) | 32 | 5820.8 |
| | Mamba2 (24+4) | 32 | 5084.8 |
| | Llama† | 32 | 3084.8 |
| | Llama (4 + 1) | 32 | 6057.6 |
| | Llama (32+4) | 32 | 7225.6 |
| | Sum | | 83132.16 |

# E. Additional Results

### E.1. Inference Dynamics and Convergence of Implicit Language Models

In Table Table 6, we present the average number of steps that the implicit language models require to process a sequence length of 2048 from the test split of the D-PILE dataset during inference in simultaneous mode. Moreover, the table also shows the relative error difference of the solutions found by the language models. Notably, the (4+1) configurations achieve a fixed point, despite not being explicitly constrained to do so. Our observations further reveal that the implicit models trained with a full DEQ setup—(24+4) for Mamba2 and (32+4) for Llama—consistently reach fixed points within their training stop thresholds, which are $< 0.05$ and below the inference step threshold of four times 24 and 32, respectively.

### E.2. Length Extrapolation Capabilities in Pretrained Models

We evaluated the ability of our models to extrapolate to longer sequence lengths and compared their performance with baseline models. All our in-house trained models, including the Mamba2 and Llama baselines, were initially trained on

sequences of 2048 tokens and subsequently tested on sequences of 4096, 8192, and up to 16384 tokens. Table 7 presents the average perplexities across different model scales. We note that the original Mamba2 models, denoted as Mamba2*, were trained on sequence lengths of 8192. Our observations indicate that in all instances, our implicit models, including the Mamba2 (4+1), and Mamba2 (24+4) as well as the Llama(32+4) configuration, maintain lower perplexities compared to the explicit Mamba2 and Llama respectively. We also found that the difference in perplexity between the longer and shorter sequences becomes more pronounced as the size of the models increases.

### E.3. Effective Duality between Simultaneous Mode and Sequential Mode in SSMs

Empirically, we demonstrate that once implicit state-space models (SSMs) are trained in a simultaneous mode (parallelizable in token dimension), these trained models can be utilized in sequential mode. In sequential mode, self-iteration occurs for each token, thereby affirming the duality of the two modes. We employ the 1.3B Mamba2 (24+4) model and 1.3B Llama (32+4) model, trained on 207B tokens from the D-PILE, to evaluate the duality between the two modes using examples from the test split of the D-PILE. Table 8 presents some detokenized outputs of the model in both modes.

Table 5: Sample types from the Catbabi (Schlag et al., 2021) dataset, a modified version of the bAbI dataset, categorized into 20 examples. These examples are derived from (Weston et al., 2015).

---

**Task 1: Single supporting fact**

sandra went back to the hallway . john moved to the hallway . where is sandra? **hallway** john travelled to the bathroom . daniel travelled to the office . where is daniel? **office** john moved to the office . sandra travelled to the bathroom . where is sandra? **bathroom** daniel went back to the bedroom . daniel went back to the garden . where is daniel? **garden** sandra moved to the hallway . john went back to the bathroom . where is daniel? **garden** .

---

**Task 2: Two supporting fact**

mary went back to the bathroom . john went to the office . daniel grabbed the football there . sandra travelled to the bathroom . daniel left the football . sandra moved to the bedroom . sandra journeyed to the kitchen . sandra travelled to the garden . sandra went back to the bathroom . john travelled to the kitchen . daniel moved to the garden . sandra moved to the garden . mary went to the office . daniel went to the kitchen .

---

**Task 3: Three supporting fact**

john went back to the bedroom . sandra got the milk . sandra discarded the milk . sandra took the milk . john went back to the office . mary moved to the bathroom . john travelled to the kitchen . daniel went back to the bedroom . john went back to the office . mary took the apple . mary travelled to the office . daniel went back to the kitchen . mary moved to the bathroom . sandra dropped the milk there . where was the apple before the bathroom ? **office**

---

**Task 4: Two argument relations**

the bedroom is west of the bathroom . the kitchen is west of the bedroom . what is west of the bathroom ? **bedroom**

---

**Task 5: Three argument relations**

jeff journeyed to the garden . fred went to the office . fred went to the hallway . fred got the apple there . fred moved to the office . fred went to the kitchen . fred put down the apple . fred took the apple there . mary went back to the garden . fred travelled to the bathroom . jeff grabbed the milk there . bill went to the office . jeff journeyed to the bathroom . jeff put down the milk there . fred picked up the milk there . fred passed the apple to jeff . who gave the apple to jeff ? **fred**

---

**Task 6: Yes/No questions**

john went to the office . sandra went to the bathroom . is sandra in the kitchen ? **no** sandra travelled to the garden . sandra went to the bedroom . is sandra in the bedroom ? **yes** sandra travelled to the garden . mary went to the kitchen . is sandra in the garden ? **yes** john grabbed the apple there . daniel journeyed to the bedroom . is sandra in the bedroom ? **no** john picked up the football there . john took the milk there . is mary in the bedroom ? **no**

---

**Task 7: Counting**

mary got the apple there . john went back to the kitchen . how many objects is mary carrying ? **one** john moved to the garden . mary left the apple there . how many objects is mary carrying ? **none** john moved to the bathroom . mary grabbed the apple there . how many objects is mary carrying ? **one** john went to the garden . mary dropped the apple . how many objects is mary carrying ? **none** sandra journeyed to the bedroom . mary moved to the kitchen . how many objects is mary carrying ? **none**

---

**Task 8: List/Sets**

john went to the bathroom . daniel travelled to the garden . mary moved to the hallway . sandra moved to the bedroom . daniel journeyed to the hallway . mary picked up the apple there . what is mary carrying ? **apple** sandra travelled to the bathroom . mary dropped the apple . what is mary carrying ? **daniel** moved to the office . daniel grabbed the football there . what is mary carrying ? **nothing** daniel moved to the bedroom . mary got the apple there . what is daniel carrying ? **football**

---

**Task 9: Simple negation**

daniel is in the bathroom . sandra is no longer in the bedroom . is daniel in the kitchen ? **no** mary is no longer in the bedroom . sandra is no longer in the kitchen . is sandra in the kitchen ? **no** sandra is in the hallway . daniel is in the garden . is sandra in the hallway ? **yes** sandra is in the bedroom . sandra is in the bathroom . is sandra in the bedroom ? **no** sandra travelled to the hallway . john is in the kitchen . is sandra in the kitchen ? **no**

---

**Task 10: Indefinite knowledge**

julie is either in the bedroom or the cinema . bill journeyed to the cinema . is bill in the cinema ? **yes** bill is in the office . julie went to the school . is julie in the park ? **no** julie travelled to the office . mary is in the school . is bill in the park ? **no** mary is in the park . bill is either in the office or the kitchen . is mary in the kitchen ? **no** fred travelled to the office . mary went back to the kitchen . is mary in the kitchen ? **yes**

---

**Task 11: Basic conference**

daniel journeyed to the garden . afterwards he journeyed to the kitchen . where is daniel ? **kitchen** sandra went back to the kitchen . then she moved to the office . where is daniel ? **kitchen** john moved to the office . afterwards he journeyed to the bedroom . where is john ? **bedroom** daniel journeyed to the office . after that he moved to the hallway . where is john ? **bedroom** sandra went to the hallway . following that she journeyed to the bathroom . where is daniel ? **hallway**

---

**Task 12: Conjugation**

daniel and mary journeyed to the bedroom . mary and sandra moved to the bathroom . where is daniel ? **bedroom** sandra and john journeyed to the hallway . mary and sandra went back to the bedroom . where is sandra ? **bedroom** sandra and mary went to the office . sandra and daniel went back to the bathroom . where is sandra ? **bathroom** mary and daniel went to the bedroom . john and mary travelled to the bathroom . where is john ? **bathroom** mary and daniel went to the hallway . john and daniel went back to the bedroom . where is daniel ? **bedroom**

---

**Task 13: Compound conference**

sandra and daniel journeyed to the bedroom . following that they travelled to the kitchen . where is daniel ? **kitchen** john and mary travelled to the bathroom . then they went to the hallway . where is john ? **hallway** daniel and sandra travelled to the office . following that they went to the bathroom . where is sandra ? **bathroom** daniel and sandra moved to the kitchen . then they went to the bathroom . where is sandra ? **bathroom** mary and john went to the kitchen . then they went to the bedroom . where is john ? **bedroom**

---

**Task 14: Time reasoning**

bill travelled to the office yesterday . bill moved to the school this morning . fred went back to the kitchen yesterday . julie journeyed to the school yesterday . where was bill before the school ? **office** this afternoon bill journeyed to the cinema . yesterday mary journeyed to the park . where was bill before the cinema ? **school** fred journeyed to the bedroom this morning . julie went back to the kitchen this morning . where was bill before the cinema ? **school** fred travelled to the park this afternoon .

---

**Task 15: Basic deduction**

mice are afraid of cats . cats are afraid of wolves . emily is a cat . wolves are afraid of cats . jessica is a mouse . gertrude is a wolf . sheep are afraid of mice . winona is a wolf . what is winona afraid of ? **cat** what is jessica afraid of ? **cat** what is jessica afraid of ? cat what is jessica afraid of ? **cat**

---

**Task 16: Basic induction**

greg is a swan . bernhard is a rhino . julius is a frog . bernhard is white . brian is a rhino . julius is green . greg is white . brian is green . lily is a swan . what color is lily ? **white**

---

**Task 17: Positional reasoning**

the red square is to the right of the triangle . the pink rectangle is below the red square . is the triangle to the left of the pink rectangle ? **yes** is the triangle to the right of the pink rectangle ? **no** is the triangle below the pink rectangle ? **no** is the triangle above the pink rectangle ? **yes** is the pink rectangle to the right of the triangle ? **yes** is the pink rectangle below the triangle ? **yes** is the triangle to the right of the pink rectangle ? **no** is the pink rectangle to the right of the triangle ? **yes**

---

**Task 18: Size reasoning**

the chocolate fits inside the container . the box is bigger than the chest . the chest fits inside the container . the box of chocolates fits inside the container . the chest is bigger than the suitcase . is the suitcase bigger than the box ? **no** does the box fit in the suitcase ? **no** does the box fit in the suitcase ? **no** is the suitcase bigger than the box ? **no** is the suitcase bigger than the box ? **no**

---

**Task 19: Path finding**

the bathroom is south of the garden . the kitchen is north of the bedroom . the hallway is north of the garden . the bedroom is west of the garden . the office is west of the hallway . how do you go from the hallway to the bedroom ? **s,w**

---

**Task 20: Agent's motivation**

yann is tired . where will yann go ? **bedroom** jason is bored . where will jason go ? **garden** antoine is bored . where will antoine go ? **garden** antoine moved to the garden . why did antoine go to the garden ? **bored** antoine got the football there . why did antoine get the football ? **bored** sumit is hungry . where will sumit go ? **kitchen** sumit travelled to the kitchen . why did sumit go to the kitchen ? **hungry** yann travelled to the bedroom . why did yann go to the bedroom ? **tired**

Table 6: Inference dynamics and convergence characteristics of implicit language models across different scales. We show the performance of Mamba2 and Llama models at various parameter scales—130M, 350M, 760M, and 1.3B—when processing sequences of length 2048 from the test split of the Pile dataset. The average number of steps required for convergence during inference in simultaneous mode is detailed alongside the relative error difference of the solutions obtained by the models. Remarkably, the (4+1) configurations reach a fixed point organically, without explicit constraints enforcing this behavior. The table further highlights that implicit models employing a full DEQ setup—(24+4) for Mamba2 and (32+4) for Llama—demonstrate consistent convergence within their predefined stop thresholds, which are less than 0.05. These thresholds are also below the designated inference step threshold of four times 24 and 32 for the respective models.

|  | Model | D-Pile Perplexity | Inference Steps | Rel. Diff. |
|---|---|---|---|---|
| 130M | Mamba2(4+1) | 13.76 | 14 | 0.035 |
| | Mamba2(24+4) | 12.86 | 62 | 0.036 |
| | Llama(4+1) | 12.73 | 13 | 0.014 |
| | Llama(32+4) | 11.73 | 53 | 0.033 |
| 350M | Mamba2(4+1) | 10.02 | 10 | 0.012 |
| | Mamba2(24+4) | 9.70 | 49 | 0.024 |
| | Llama(4+1) | 9.66 | 12 | 0.015 |
| | Llama(32+4) | 9.43 | 57 | 0.037 |
| 760M | Mamba2(4+1) | 8.60 | 10 | 0.013 |
| | Mamba2(24+4) | 8.35 | 45 | 0.025 |
| | Llama(4+1) | 8.27 | 12 | 0.014 |
| | Llama(32+4) | 7.90 | 77 | 0.044 |
| 1.3B | Mamba2(4+1) | 7.97 | 10 | 0.013 |
| | Mamba2(24+4) | 7.70 | 47 | 0.029 |
| | Llama(4+1) | 7.66 | 13 | 0.015 |
| | Llama(32+4) | 7.24 | 69 | 0.048 |

Table 7: Length extrapolation performance of pretrained models on varying sequence lengths. We demonstrate the average Pile test perplexity results for Mamba2 (300B tokens) and our in-house trained Mamba2* (207B tokens) baseline models, as well as our Mamba2(4+1) and Mamba2(24+4) configurations, across a range of sequence lengths—2048, 4096, 8192, and 16384 tokens. Each model was originally trained on sequences of 2048 tokens, while the original Mamba2 was trained on 8192 tokens. The results highlight that our implicit models consistently achieve lower perplexities than their explicit counterparts across all tested sequence lengths. Notably, as model sizes increase, the discrepancy in perplexity between longer and shorter sequences grows more evident.

|  | Model | 2048 ppl↓ | 4096 ppl↓ | 8192 ppl↓ | 16384 ppl↓ |
|---|---|---|---|---|---|
| 130M | Mamba2 | 13.72 | 13.44 | 13.92 | 14.91 |
| | Mamba2* | 13.05 | 12.72 | 12.76 | 12.94 |
| | Mamba2(4+1)-ours | 13.76 | 13.04 | 13.06 | 13.25 |
| | Mamba2(24+4)-ours | **12.86** | **12.53** | **12.55** | **12.78** |
| 370M | Mamba2 | 10.55 | 10.28 | 11.20 | 13.89 |
| | Mamba2* | 10.18 | 9.94 | 10.07 | 10.34 |
| | Mamba2(4+1)-ours | 10.02 | 9.74 | 9.73 | 9.86 |
| | Mamba2(24+4)-ours | **9.70** | **9.45** | **9.44** | **9.59** |
| 760M | Mamba2 | 9.23 | 9.45 | 34.99 | 231.25 |
| | Mamba2* | 8.98 | 8.94 | 10.38 | 12.56 |
| | Mamba2(4+1)-ours | 8.60 | 8.38 | 8.40 | 8.55 |
| | Mamba2(24+4)-ours | **8.35** | **8.16** | **8.18** | **8.33** |
| 1.3B | Mamba2 | 8.40 | 8.47 | 12.69 | 25.37 |
| | Mamba2* | 8.28 | 8.24 | 9.96 | 15.06 |
| | Mamba2(4+1)-ours | 7.97 | 7.81 | 8.36 | 9.53 |
| | Mamba2(24+4)-ours | **7.70** | **7.59** | **8.19** | **9.63** |

Table 8: Selected examples from the test split of the D-Pile dataset, showcasing the detokenized output of the next-token predictions by the 1.3B implicit Mamba2 model, which was trained in a parallel fashion and tested in both Simultaneous and Sequential modes. These outputs highlight the duality of the two approaches.

| Ground Truth | Simultaneous | Sequential |
|---|---|---|
| labour productivity and output would rise as a result • it is essential to protect the professionalism of certain categories of workers: the debates here centred on performance artists and female theatrical employees engaged in highly physical and intensely emotional work • heavy physical labour and strenuous exercise can lead to disruptions of the menstrual cycle • women's physical and intellectual capacities are reduced during menstruation; women lose muscular strength and powers of concentration • women's biological constitution and reproductive functions require specific recognition in law Against the provision: • employers are less likely to appoint women if they are guaranteed paid time off work during menstruation | , market. the. be. a result. The would a to ensure the interests and of the professions of workers professions on arered on the management, the artists performers in the skilled work demanding competitive work • the industry work is therenuous physical are be to injuryions in the body cycle and• the's bodies and emotional capacities are not by pregnancyation this are their strength and stam of endurance • menstru's menstrual clocks is menstrual capacity are special protection the •Thest this background of • the should not likely to be women to they are not the matern off for for menstruation • womenin un the commentators) | . market. the. be. a result of The would a to ensure the environment and of the professions of workers professions on arered on the management, the artists performers in the skilled work demanding demanding work • the industry work is therenuous physical are be to injuryions in the body cycle and• the's bodies and emotional capacities are not by pregnancyation this are their strength and stam of endurance • menstru's menstrual clocks is menstrual capacity are special protection the •Thest this background of • the should not likely to be women to they are not the matern off for for menstruation • womenin un the commentators) |
| form of "photon counting".. "This de-excitation is called 'fluorescence', and it is characterized by a lifetime of a few nanoseconds of the lowest vibrational level of the first excited state. De-excitation from the excited singlet state to the ground state also occurs by other mechanisms, such as non-radiant thermal decay or 'phosphorescence'. In the latter case, the chromophore undergoes a forbidden transition from the excited singlet state into the triplet state (intersystem crossing, ISC, Fig 2.4), which has a non-zero probability, for example because of spin orbit coupling of the electrons' magnetic moments" its a type of INTERSYSTEM CROSSING doing a search for Intersystem crossing, memristor brings up this link.. A composite optical microcavity, in which nitrogen vacancy (NV) centers in a diamond nanopillar | meaning of thethe"" is IThe isceptfactciting of a thephotonorescence' and it is the by the photonphotonetime of the few hundredoseconds. the excited- level of the excited of -excitation is the first state state is the ground state is occurs, , as -radiative , fluorescence' the case case, the excitedophore is a non transition to the ground singlet state to the ground state,'ystem crossing), ISC), or.).1). which is a lifetime-ne probability of but example, of the- coupling to the excited to spin moment. " a bit of fluorescenceC crossingOSSING, " a google on "tersystem crossing I Ieor, up a " mem material devicecavity is consisting which the- (NV) centers are diamond diamond crystalillar are coupled to aing gallery modes ( a silicon microsphere, is fabricated. | meaning of theto"" is IThe isceptfactciting of a thephotonorescence' and it is the by the photonphotonetime of the few hundredoseconds. the excited energy state of the excited of -excitation is the first state state is the ground state is occurs, , as -radiative , fluorescence' the case case, the excitedophore is a non transition to the ground singlet state to the ground state,'ystem crossing), ISC), or.).1). which is a lifetime-ne probability of but example, of the- coupling to the excited to spin moments. " a bit of fluorescenceC crossingOSSING, " a google on "tersystem crossing, Ieor, up a " mem material devicecavity is consisting which the- (NV) centers are diamond diamond crystalillar are coupled to aing gallery modes ( a silicon microsphere, is fabricated. |

