# OpenReview forum: "Implicit Language Models are RNNs: Balancing Parallelization and Expressivity"
_ICML.cc/2025/Conference — ICML 2025 spotlightposter_

### Official Review · Reviewer_e8CX · 2025-02-24

**Overall Recommendation:** 4

**Summary:**

This paper introduces implicit SSMs, which are a parameter tied form of SSM that can be run for arbitrarily many self-iterations until convergence. They propose training implicit SSMs in a scalable manner using phantom gradients from the implicit layers literature. They demonstrate the ability of implicit SSMs to state track on hard OOD settings and in language modeling.

**Claims And Evidence:**

* The discussion around equations (6) and (7) implies that these equations converge to a fixed-point. However, is this correct? Not all iterative equations converge, and no proof of this claim is given.
* > Theoretically, we show that implicit SSMs implement the non-linear state transitions of RNNs
    * This claim is formalized in Theorem 1
    * I am concerned about the shift from $h_t^*$ to $h_{t-1}^*$ in the proof, see my comments in the "Theoretical Claims" section
* > Empirically, we find that only approximate fixed-point convergence suffices
    * Figure3, Mid supports this claim
* Lines 1080-1 (Figure 10 caption)
> The implicit Mamba2 retains its performance as the story length increases, whereas the explicit Mamba2's performance declines
    * This claim is contradicted by the performance of the explicit 3 layer in Figure 10c, which on the balance appears to have better performance as story length increases.
* Lines 331-4
> The implicit Mamba2 models maintain their perplexity as sequence length increases, whereas the baseline Mamba2 models exhibit an increase in perplexity with longer sequences
    * Yes, demonstrated in Figure 4
* Effective Duality between Simultaneous Mode and Sequential Mode
    * Yes, this claim is supported

**Essential References Not Discussed:**

The authors cite Lim et al '24, but note that their method incurs cubic cost in terms of state size. The authors may also wish to cite
* Gonzalez et al '24, "Towards Scalable and Stable Parallelization of Nonlinear RNNs," https://arxiv.org/abs/2407.19115
which is an extension of Lim et al but uses quasi-Newton methods to avoid the cubic cost in state size.

The authors may also consider citing the following paper which started the deep SSM line of work

* Gu et al '21, "Combining Recurrent, Convolutional, and Continuous-time Models with Linear State-Space Layers," https://arxiv.org/abs/2110.13985

In particular, see Appendix C.

In this paper, Gu et al prove that many layers of an SSM can approximate a Picard iteration (a fixed point iteration). The methods in this proposed paper are effectively doing a Picard iteration as I understand it, so some comparison with the theory developed by Gu et al '21 may be useful to the academic community.

While not required because of the ICML policy on concurrent work, the authors might considering citing and discussing the concurrent work
* Geiping et al '25, "Scaling up Test-Time Compute with Latent Reasoning: A Recurrent Depth Approach." https://arxiv.org/abs/2502.05171
They also uses self-iterations, but with attention layers instead of SSM layers. They also apply to language modeling. A robust discussion of similarities and differences in approach would be a great resource for the community.

**Experimental Designs Or Analyses:**

* Will the code be published? It is difficult to check the experiments without code.
* How was $\lambda$ set for the Phantom gradients (see equation 5)? This choice of hyperparameter does not seem to be discussed anywhere in the paper, even though there is a very good treatment of other experimental design choices. What happens if this hyperparameter $\lambda$ is varied?
* In Figure3 Right, I am concerned that the explicit models did the worst on train accuracy. Is there an explanation for this behavior? I would have thought that 16 layers of mamba would be enough to memorize to memorize a sequence length of length 256, i.e. get to at least 90% train accuracy. Is there any way to explain this phenomenon or provide evidence that the explicit models are being trained to the utmost on the synthetic state tracking task (Figure 3).
* I would really like to see more reporting of wall clock time and memory usage in the experiments. I discuss wall clock time at length in "Other Strengths and Weaknesses" Section. As for memory, I was extremely impressed by the batch size of 1M tokens for the language modeling experiments. I want to know how much memory was required for training with a batch size of 1M tokens. Such an addition of max memory needed for training should be added to Table 6.
* Broadly speaking on the Language Modeling experiments (Table 1, Table 5, etc), I wasn't sure if a proper ablation was done for depth. For example, what would happen if the explicit models were made deeper but less wide (to preserve parameter matching). Their depth could be scaled by the number of inference steps reported in Table 2 (are those inference  steps means over tokens in Table 2?).
* The large scale language modeling tasks in Table 1 are good and show modest improvement of implicit over explicit models, but they do not blow away the explicit models. Are there any tradeoffs, i.e. downsides of using an implicit model, in terms of memory, compute, wallclock time, or some other metric? An explicit bolded paragraph on "Limitations" would be a nice contribution to contextualize the method and help practitioners.

**Methods And Evaluation Criteria:**

Yes, the proposed methods and evaluation are excellent.
* The OOD task in Figure3 was a great addition to the literature
* The use of the CatbAbI dataset was a good idea

**Other Comments Or Suggestions:**

Style
* The capitalization in this paper is a bit nonstandard at times, i.e. sometimes too generous with capitalization (eg "Illusion of State", "Word Problem", "Implicit Function Theorem"; also capitalization of a sentence fragment after a colon), but then other times doesn't capitalize "theorem" when it should. The authors may wish to review standard English capitalization style guides.

Typos
* In equation 2, I think it should be $h_t$ and not $h_{t-1}$
* Line 434: the quotes around 'hard' are a bit ugly.
* Line 662: "it's" should be "its"
* Line 699
     >  Monoid whose elements can be inverted have a particularly right structure
    * Perhaps this sentence should read: "Monoids where every element has an inverse are called \emph{groups}."

Small Suggestions
* Another nit, but in lines 199-202, the authors write
> The Illusion of State reveals that SSMs cannot simulate arbitrary finite state machines
    * wouldn't it be better style to not capitalize illusion of state?
    * I think SSMs can in fact simulate arbitrary finite state machines, just not without depth growing in the sequence length. I think the sentence should instead read "\citet{Merrill24} shows that SSMs cannot simulate arbitrary finite state machines with constant depth."
* Another nit: line 202-4
> A hard state tracking problem in the sense that all state tracking problems can be reduced to it
    * not the most elegantly phrased sentence
    * strictly speaking, not _all_ state tracking problems can be reduced to $S_5$ (consider for example $\mathbb{Z}_7$).

**Other Strengths And Weaknesses:**

On the balance, I think this is a great paper. However, I have various minor concerns scattered throughout this review that I think should be addressed. More importantly, I have a major concern about **wall-clock time** that I elaborate on in this section. I really need to see this question of wallclock time directly addressed before I can advocate for publication.

_Synthetic State tracking task (Figure 3)_

For example, in all of Figure 3, the implicit models are granted unbounded test iterations (what actually is the halting condition? I do not think the halting condition is stated explicitly in the paper).

What is the wall-clock time for inference of the implicit models at test time in Figure3, compared to the wall clock time of Mamba2 (explicit) for inference?

Also, I'm curious about the fairness of the Mamba2 baseline. In Figure3 right, the explicit and implicit models have matched train time depth, which is very good. But what happens if we match test time depth? I.e., report the average number of test time iterations used by implicit mamba in Figure3 right. And then give that many layers in depth for both training and test to explicit Mamba2 (go ahead and parameter match still). **I would really like to see this fair baseline for Mamba2 before concluding that explicit layers are limited on this task.**

However, this OOD task with increasing the number of $S_5$ tokens was extremely clever and a great addition to the literature.

_catbAbI task (Appendix D.2)_

What is the wall clock time, both for training and for test, of the implicit and explicit mamba2 models (1,2, and 3 layers) on the catbAbI dataset. As demonstrated in Figure10a, a 3 layer explicit mamba2 has almost identical performance to a 1 layer implicit mamba2. However, if implicit mamba2 takes dramatically longer on wallclock time, it is difficult to recommend the use of 1 layer implicit (with a large number of fixed-point iterations) over a 3 layer explicit mama2.

How should I reconcile Figure 10a and Figure 10c however? On 10a it looks like explicit and implicit 3 layers almost always have similar performance; while on Figure10c it looks like implicit 3 layer is always better than explicit 3 layer. But how would a parameter matched explicit 6 layer do in comparison, both in test accuracy and on **training and test time wall clock time.**

_Language Modeling task_

It would be helpful to add wallclock time (for both training and inference) to Table 6 (in addition to peak memory requirement for training and inference, see discussion in "Experimental Designs or Analyses.")

_Extrapolation Advantage_

A main selling point of this paper (especially on the synthetic state tracking experiment and on the length extrapolation aspect of language modeling) is that implicit models seem to be better on extrapolation tasks (proportion of hard tokens, length of sequence) at test time. **Can you provide any theoretical perspective on why implicit models are better at extrapolation?** Doing so would really strengthen the paper!

**Questions For Authors:**

1. How is Figure 2B related to equation (8)? It would seem that every token in depth still needs to propagate across the sequence length in order to simulate equations (6) and (7), which converge to (8). Why is only $h_t^*$ needed to be passed on, instead of all $h_t^{(s)}$?
2. Would you please elaborate on the following sentence in your conclusion?
> While implicit models lift the limitations of state-of-the-art language models, self-iteration comes at a cost that only amortizes over the long tail of natural language
3. In Figure 3 (Left and Mid), there is a large discrepancy between the best run and all runs. Would you please elaborate on what is causing this difference? Is there any way to see from train performance alone which run will yield the highest test accuracy? Also, how many self-iterations are used on average at test time in Figure 3 (all 3 panels)? And how many self-iterations are used during training for Figure 3 Left?
4. Could more detail be provided about the following sentence on lines 251-5?
> Interestingly, the number of test time self-iterations is quite similar for the models trained with different upper bounds on the training time self-iterations, hinting that the models learn similar algorithms
How many test time self-iterations is this?
5. Why in Figure3 Right does unrolled have better training performance than phantom gradients, but the opposite relation is true for test?
6. For Figure3 Right, an important missing baseline would seem to be a truncated backpropogation, i.e. only backpropogate through 8 steps (as is done in Geiping et al 2025, ""Scaling up Test-Time Compute with Latent Reasoning", https://arxiv.org/abs/2502.05171). This approach may be somewhat intermediate in the train/test tradeoff between phantom gradients and full unrolled backpropogation, and would also be constant memory. I think you are already doing this on the CatbAbi task, see lines 927-8.
7. In the catbAbi experiments (D.2), did you ever try an ablation where you unrolled the entire way through, and never switch to self-iteration fixed-point search? In Figure 8, there is a sharp dip in validation accuracy at 5000 steps when you switch schedules;  but otherwise, it doesn't look like the validation trajectory was changed very much as a result of the change in training procedure. Moreover, I thought that the point of Figure 3 Mid was that fixed number of self-iterations at train time was an acceptable way to train the model.
8. I must not be understanding something about Figure 9 (maybe the y-axis should be renamed from "Validation steps" to "number of self-iterations"). Still, I thought that based on lines 927-8 that 32 self-iterations would be used on the first 5000 gradients steps, but it looks more like 4. How was this number chosen? And was does "trained for 5000 steps in unrolling mode, utilizing 32 steps with normal gradient checkpointing" mean?

**Relation To Broader Scientific Literature:**

This paper builds on the implicit layers line of work to create an implicit model that can actually perform well on language tasks!

**Theoretical Claims:**

_Theorem 1 and Proof in Appendix B_

* This is a bit of a nit / more of a notational question, but in equation (14), wouldn't it be more correct to write $\dfrac{d h_t^*}{d h_{t-1}^*}$ (as opposed to $\dfrac{\partial h_t^*}{\partial h_{t-1}^*}$, as currently written)? Looking at equation (13), it seems to me like $\dfrac{\partial h_t^*}{\partial h_{t-1}^*} = \Lambda(z_t^*, x_t)$, while the full derivative contains the off diagonal correction terms.
* **A more important point that needs to be discussed more precisely and clearly** is the choice of $h_{t-1}^*$ as opposed to $h_t^*$ as the argument of $\varphi$. In particular, on line 785, it is stated without justification that $z_t^* = \varphi(h_{t-1}^*, x_t, \theta)$. However, in equation (7), $z_t^{(s)}$ is a function of $h_t^{(s)}$ and not of $h_{t-1}^{(s)}$. This distinction between $h_{t-1}^*$ as opposed to $h_t^*$ is important because if $\varphi$ is actually a function of $h_t^*$, then the off-diagonal terms in equation (14) disappear and the Theorem is incorrect as stated. Therefore, I think it is very important for the authors to **add a lot more rigorous detail** about why $\varphi$ takes $h_{t-1}^*$ as an argument instead of $h_t^*$, especially because this shift is different from the set up for equations (6) and (7)
    * I also think it is very important that the authors provide a numerical check of equation (14) in their trained models. I.e, in their trained models, what actually are the Jacobians $\dfrac{\partial h_t^*}{\partial h_{t-1}^*}$. Do they correspond within numerical tolerance to the RHS of equation (14)? Or not? Please include such a numerical check in your rebuttal.

_Phantom Gradient (Section 2.3)_
Is the minus sign in equation (4) correct?
As I understand things, we know that
$$ G(\Phi, x, \theta) = 0.$$

Thus, taking derivatives wrt $\theta$, it follows that
$$ \dfrac{\partial G}{\partial \theta} + \dfrac{\partial G}{\partial z} \dfrac{\partial \Phi}{\partial \theta} = 0.$$

Now, we know that $\dfrac{\partial G}{\partial \theta} = - \dfrac{\partial F}{\partial \theta}$, and so plugging in it follows that
$$   \dfrac{\partial G}{\partial z} \dfrac{\partial \Phi}{\partial \theta} =  \dfrac{\partial F}{\partial \theta}.$$

Therefore, it would seem that there is a sign error in equation 4.

---

> ### Author Rebuttal · Authors · 2025-03-31
>
> # General comment
> We thank the reviewer for lots of insightful feedback, which significantly helps us to revise our manuscript. We appreciate the positive evaluation ("On balance, I think this is a great paper") as well as the engagement expressed by many detailed questions. In face of the space limit (5k characters), we have to focus on selected questions.
> # Theoretical contributions
> While we are not aware of theoretical convergence guarantees, we checked that our models converge to fixed points on all datasets that we present in the paper. Models that reach fixed points have the properties claimed in Section 3.
>
> > $h^*_{t-1}$ vs $h^*_t$ in $\varphi$.
>
> There is a typo in Eq. (7), which should be analogous to Eq. (2), where the dependency on $h_{t-1}$ is correctly stated in line with standard RNN formulations. Consequentially $\varphi$ takes $h_{t-1}^*$ as an argument.
>
> > Is the minus sign in equation (4) correct?
>
> No, thanks for spotting this!
>
> > Please include such a numerical check in your rebuttal.
>
> Incorporating the correct sign, we compared the RHS of Eq. (14) at the fixed point with the unrolled autograd (AD) Jacobian. The absolute difference between the AD Jacobian and Eq. (14) is three orders of magnitude smaller than the values in the Jacobian.
>
> # Addressing wall-clock time
> We have included the WCT for all datasets and the memory of language models in our response to __AqCH__.
> ## Synthetic state tracking
> The halting condition is convergence of $z$ (relative diff. of 5%).
> - setting an upper limit of 4 self-iterations takes 26s on the test set (4k examples) and gets 97.8% accuracy on the p=0.5 distribution.
> - an upper limit of 16 self-iterations takes 35s and gets 99.8%. Note that iterations terminate after 6 steps on average due to the above halting condition.
> - the explicit model with 16 layers takes 37s and gets 1.5%.
> We believe that this is a fair baseline.
>
> > this OOD task with increasing the number of $S_5$ tokens was extremely clever and a great addition to the literature.
>
> thanks!
> ## catbAbI
> The 1-layer implicit model trains faster than the 3-layer explicit model, but takes slightly more time during inference (see  __AqCH__).
>
> The discrepancy between Figures 10a and 10c arises because Figure 10a averages accuracy across all story lengths per task, obscuring differences related to the distribution of story lengths that varies a lot across tasks and is also non-uniform per task.
>
> > how would a parameter matched explicit 6 layer do in comparison
>
> According to Fig. 1, a deep enough model will sufficiently track state, which should also hold for catbAbI. We view catbAbI as an intermediate sanity check between our synthetic task and the larger language models, and hence did not investigate this further.
>
> ## Language Modeling
> We will add the values reported to __AqCH__ to table 1 and table 6.
> # Questions for authors
> Q1: It is an empirical contribution of this work that passing on $h_t^*$ suffices for inference, which enables const memory language generation. Intuitively, DEQs are all about fixed points, and “path independence” has been observed in prior works.
>
> Q2: Self-iteration introduces additional FLOPs. Many practical problems might already be sufficiently addressed by the baseline models. Similar to test-time computation, self-iteration pays off only for certain problems.
>
> Q3: There are multiple runs that  overlap with the best run. Perhaps a box-plot would be a more appropriate choice. Left was trained with 32 iterations.
>
> Q4:
> about 6 iterations (5-7 for different models) per token.
>
> Q5:
> It seems that differentiating locally around the fixed point and not along the full trajectory provides a stronger bias towards learning sequential problems.
>
> Q6:
> Phantom gradients are similar to the truncated backpropagation method used by Geiping et al., but uses the update rule $z_{t+1} = (1 - \lambda) z_t + \lambda f(z_t, x)$. For $\lambda=1$, the two algorithms match.
>
> Q7:
> We found that continuing the 4x unrolling for the complete training is not enough to achieve competitive accuracy.
>
> Q8:
> '32' is indeed a typo. Should be 4x unrolling till step 5000.
>
> # Further comments
> As requested by multiple reviewers, we will add a new section on limitations to the manuscript to discuss the moderate downstream task improvements in face of the larger wall-clock time.
>
> > Can you provide any theoretical perspective on why implicit models are better at extrapolation?
>
> Prior works indicate that implicit models have higher robustness to noise due to attractor dynamics. Viewing uninformative tokens as a source of noise (e.g. at longer sequence length) might explain the robustness.
>
> > provide evidence that the explicit models are being trained to the utmost on the synthetic state tracking task
>
> Fig. 1 (top left) shows that the number of layers required by the explicit model to solve the S5 problem grows linearly with the sequence length showing that explicit models are limited by their depth.

---

> > ### Comment · Reviewer_e8CX · 2025-04-02
> >
> > Thank you very much for your response. Your response to reviewer AqCH regarding memory and wallclock time is excellent and should definitely be included in your final paper. I think the explicit limitations section will be a great addition as well.
> >
> > In light of the limitations regarding memory and wallclock time, what would you say the practical benefits of the implicit SSM model are, against deeper but less wide explicit models?
> >
> > Would it also be possible to answer these two of my original questions?
> >
> > * Will the code be published?
> > * How was $\lambda$ set for the Phantom gradients (see equation 5)? This choice of hyperparameter does not seem to be discussed anywhere in the paper, even though there is a very good treatment of other experimental design choices. What happens if this hyperparameter is varied?
> >
> > One more question:
> >
> > > Q1: Intuitively, DEQs are all about fixed points, and “path independence” has been observed in prior works.
> >
> > What prior works regarding "path independence" are you referring to? Could you emphasize this point (including citations) more in the main text?
> >
> > With the addition of the wallclock time experiments, this is an excellent paper that is way too interesting not to publish. I am raising my score to a 4. I really hope this paper gets in. I'm still not sure if the method is practically useful however, and would be interested in a candid discussion from the authors.

---

> > > ### Author Response · Authors · 2025-04-04
> > >
> > > We highly appreciate the reviewers engagement and their perception of our work. We will include the additional information requested by the reviewers in the main text where possible, and will provide a complete overview in the appendix.
> > >
> > > >In light of the limitations regarding memory and wallclock time, what would you say the practical benefits of the implicit SSM model are, against deeper but less wide explicit models?
> > >
> > > The ratio between depth and width does not seem to fundamentally affect the performance of language models (https://arxiv.org/abs/2001.08361).
> > > Fig. 3 (right) shows that explicit models trained at the same depth do not generalize to harder samples or longer sequences, despite achieving comparable training accuracy to implicit models. This shows that there exist problems (e.g. S5 word problem) where implicit models are able to capture the intrinsic algorithm, and explicit models trained with the same depth are not able to capture the intrinsic algorithm.
> > >
> > > > I'm still not sure if the method is practically useful however, and would be interested in a candid discussion from the authors.
> > >
> > > We acknowledge that problems where explicit models fail to capture the intrinsic algorithm might occur only rarely in natural language . While many tasks for chat-assistants might be perfectly well addressed with state-of-the-art models, problems like analyzing and completing code, architecting software, static program analysis, or sequence models for controlling industry processes might benefit from enriched expressivity. Recent studies uncover issues of transformers with constructing internal world models (https://arxiv.org/abs/2406.03689). The ability of implicit SSMs to implement arbitrary finite state machines could lead to improved world models, and we are excited to explore this property in future research.
> > >
> > > Furthermore, test-time computation and reasoning has been a major research direction over the past few months. Self-iteration can be viewed as reasoning in latent space which received recent interest (e.g. https://arxiv.org/abs/2412.06769). The role of implicit models in this direction has to be elaborated in future work. The structural similarity of concurrent works such as Geiping et al., 2025 suggests that our theoretical contributions hold for their model as well.
> > >
> > > We generally agree that GPUs (or any von Neumann architecture) are not a perfect match for self-iterations. However, token generation on GPUs is heavily bottlenecked by HBM memory bandwidth, leaving compute cores underutilized. We believe that there is space for optimizing token-generation in implicit models, particularly transformers, on GPUs by parallelizing the wave-front $s + t = \text{const}$, which would allow to amortize HBM memory transfer for multiple steps in the iteration by increasing the arithmetic intensity. Please note that the kv-cache here is shared in the depth and token direction (unlike batching which increases kv cache size).
> > >
> > > Recently, emerging computational paradigms such as in-memory computing eliminate the necessity of transferring model weights for every iteration, which might favor models involving self-iterations and be a perfect match for implicit SSM.
> > >
> > > > Will the code be published?
> > >
> > > Yes, the code and all experimental configurations will be published soon!
> > >
> > > > How was $\lambda$ set for the Phantom gradients (see equation 5)? […] What happens if this hyperparameter is varied?
> > >
> > > We’d like to refer to Sec 2.3, where we state that $\lambda$ “helps maintaining a small condition number at the cost of increased fixed-point iterations. We experimented with $\lambda = 0.5$ and  $\lambda = 0.8$ and observed no notable differences in task performance. It might be of practical interest as well that in addition to 4 phantom gradient steps, we experimented with up to 8 steps with a 130M language model. More PG steps proportionally increase the memory footprint, but did not proportionally improve the performance.
> > >
> > > > What prior works regarding "path independence" are you referring to? Could you emphasize this point (including citations) more in the main text?
> > >
> > > Note that the gradient of an implicit model by the IFT is independent of the fixed-point search trajectory. A study that shaped our intuition for path independence was https://arxiv.org/abs/2211.09961 . We view the “simultaneous/sequential duality” presented in Sec. 5 and Fig. 2 as an important contribution of our work. Therefore, we are happy to provide more context and emphasis in the revised manuscript.
> > >
> > > > I am raising my score to a 4. I really hope this paper gets in.
> > >
> > > We deeply appreciate the constructive feedback and the reviewer’s willingness to engage in a candid discussion of the method’s practical relevance. We’re glad to hear the raised score and will do our best to make the final version as clear and useful as possible.

---

### Official Review · Reviewer_oVp8 · 2025-03-13

**Overall Recommendation:** 3

**Summary:**

This paper describes an implicit approach to training state-space models with arbitrary depth by having the models evaluated in a fixed point and implicitly differentiating using the implicit function theorem, like DQEs. They find that on certain tasks, implicit SSMs outperform SSMs, which are unable to learn these tasks.

**Claims And Evidence:**

This paper claims “Notably, our implicit models outperform their explicit counterparts on standard benchmarks”, which is supported by Table 1, and can in fact model stateful systems, which is supported by Figure 1.

**Essential References Not Discussed:**

I am not familiar enough with the existing literature to fill this section.

**Experimental Designs Or Analyses:**

The experimental designs and analyses seem appropriate.

**Methods And Evaluation Criteria:**

Several benchmarks are used for evaluation, which all seem reasonably well-suited to the task.

**Other Comments Or Suggestions:**

Minor nits:
Equation (8) should probably be paired with an additional equation z^* = f_theta(z^*, h_t^*, x_t) in order to make the fixed-point nature of the definition clear.  Additionally, in the text immediately following this equation, “The fixed point z^*_t depends on h^*_t, and hence by equation (7) on h^*_{t-1}” I believe you intended to write “by equation (6)” because that is where the dependency with h_{t-1} is established.

216 right column: A5 should be A_5.

**Other Strengths And Weaknesses:**

All points are addressed elsewhere.

**Questions For Authors:**

All points are addressed elsewhere.

**Relation To Broader Scientific Literature:**

I am not familiar enough with the existing literature to fill this section.

**Theoretical Claims:**

Theorem 1 claims that the transition function in equation (8) is non-linear and non-diagonal. Appendix B contains the proof.

I am somewhat unfamiliar with this style of proof, so perhaps this is justified by an “almost always” that is implicit, or I am completely missing something. However, several times the compositionality of non-linear functions, non-diagonal matrices, etc is used, which does not hold in general.

For example, the line following equation 12 suggests that because (partial phi)/(partial h) = - (I - (partial f)/(partial z))^{-1} (partial f) / (partial h), it must be the case that f being nonlinear implies phi is nonlinear. However (again, I could be completely wrong), this is not guaranteed. For example, the equation f(h, z) = e^h e^z + z would have (partial phi)/(partial h) = - (1 - (e^h e^z + 1))^{-1} e^h e^z = - (e^h e^z)^{-1} e^h e^z = -1, so phi would be linear despite f being nonlinear. Obviously, this example is contrived specifically to be a counterexample, and the property holds in almost all cases, but unless there is something I am missing, it seems like this should be noted in the theorem statement.

Similar “two nonlinear functions exactly cancelling each other out”-style counterexamples could surely be found for the claim on line 785 and a similar style of argument could be made to find a counterexample to the claim on line 796 that this Jacobian is necessarily non-diagonal.

---

> ### Author Rebuttal · Authors · 2025-03-31
>
> # General comment
> We would like to thank the reviewer for their comments. We are pleased that the reviewer finds our methods, evaluation criteria, experimental design, and analysis satisfactory. We appreciate the suggested clarifications and would like to take this opportunity to elaborate further on our theoretical results.
>
>
> We rigorously derive the Jacobian of the hidden state-to-state in Eq. (14) by taking the derivative of the fixed point condition Eq. (8). Incorporating the reviewers feedback, we will add the corresponding fixed point condition for $z^*$ as well along Eq. (8) as
> $$z_t^* = f_\theta\left(z_t^*, h_{t-1}^*, x_t\right)$$
>
> If
> $M = \frac{\partial g_\theta}{\partial z}\vert_{z_t^*, h_{t-1}^*}$
> is non-singular, we can further apply the implicit function theorem to replace $\frac{\partial \varphi}{\partial h}$ in Eq. (14) with Eq. (12). While there might exist $\theta$ such that $M$ is singular, we numerically verified that $M$ is non-singular for randomly initialized networks, and we did not observe singular $M$ during our training experiments.
>
> Eq. (14) contains products of matrices. As the reviewer points out, there is no guarantee that these products will not cancel out all non-diagonal terms, which could effectively lead to a diagonal Jacobian. As an example, the probability of two random Gaussian matrices multiplying to a diagonal matrix is zero. For the case of Eq. (14), we did not provide a rigorous proof that the probability is zero. Yet, we numerically checked that the Jacobian is non-diagonal both using autograd as well as Eq. (14).
>
> To rule out any concerns about the rigor of our theoretical contribution, we suggest revising Theorem 1 to mention only what we can rigorously prove:
>
> Theorem 1:
> The Jacobian of the implicit SSM is given by Eq. (14). If
>
>  $$ \frac{\partial g_\theta}{\partial z}\vert_{z_t^*, h_{t-1}^*}$$
>
> is non-singular, then $\frac{\partial\varphi}{\partial h}$ is given by (Eq. (12)).
>
>
> Remark 2: In contrast to the explicit state-space model in Eq. (1) and (2) the implicit state-space model allows for non-linear and non-diagonal state-to-state transitions. We empirically observe that $\varphi$ is non-linear and that the Jacobian is non-diagonal.

---

### Official Review · Reviewer_AqCH · 2025-03-13

**Overall Recommendation:** 4

**Summary:**

1. The authors propose a DEQ-ified version (referred to as implicit models) of state space models like Mamba2.
2. This is motivated by the fact that the diagonal (and real) state transition matrix of these models is not expressive enough for state tracking. They show an implicit model has as a non-linear and a non-diagonal state transition matrix which is required for state tracking.
3. They test implicit Mamba2 model on the CatbAbi synthetic benchmark and on language modeling on D-PILE dataset.

**Claims And Evidence:**

For theoretical claims and issues, please see "Theoretical Claims*
For methodological claims and issues, please see "Methods And Evaluation Criteria"

**Essential References Not Discussed:**

N/A

**Experimental Designs Or Analyses:**

Please see "Methods And Evaluation Criteria"

**Methods And Evaluation Criteria:**

Overall, **I think this is a good paper**, however, the only methodological/evaluation concern I have is on the **wall-clock time of this method**. Please correct me if I am wrong, but I expect that this method's (4+1) version takes atleast 4x the number of FLOPs in the forward pass than vanilla Mamba2 and that it's (32+4) version would take 32x the number of FLOPs than vanilla Mamba2. To probe this a little more, I suggest the following experiments:

1. Can the authors provide a wall-clock time analysis for their method against vanilla Mamba-2
2. I believe the the authors have currently controlled for number of parameters; could they also do a language modeling experiment with a control on the FLOPs. I am curious if the model's superior performance (on sizes > 350M) is actually due to the increased computation rather than the ability of the model to do state tracking.


Furthermore, I think the wall-clock time becomes even more relevant as recent works like [1] have shown that Mamba2 can do state tracking if the transition matrix has complex eigenvalues. I think associative scan implemented for Mamba-1 already supports diagonal matrices with complex eigenvalues and it might be a more practically efficient solution to this problem. **Could the authors comment and contrast their method with this solution?** *NOTE: I would not hold this fact against this paper since [1] is a recent work.*


[1]: Unlocking State-Tracking in Linear RNNs Through Negative Eigenvalues. Riccardo Grazzi, Julien Siems, Jörg K.H. Franke, Arber Zela, Frank Hutter, Massimiliano Pontil

**Other Comments Or Suggestions:**

N/A

Running the model on more state tracking tasks like Parity and Arithmetic Mod (w or w/o brackets) might be help strengthen the paper.

[1]: Unlocking State-Tracking in Linear RNNs Through Negative Eigenvalues. Riccardo Grazzi, Julien Siems, Jörg K.H. Franke, Arber Zela, Frank Hutter, Massimiliano Pontil

**Other Strengths And Weaknesses:**

N/A

**Questions For Authors:**

Please see "Theoretical Claims" and "Methods And Evaluation Criteria"

**Relation To Broader Scientific Literature:**

Tries to fix the problem that SSMs cannot do state tracking which is known in the community.

**Theoretical Claims:**

In light of my comment on [1] in "Methods And Evaluation Criteria", I am curious if the authors can compare/comment on the difference in expressivity of their method and an SSM with a complex valued diagonal transition matrix. Is it possible to get a characterization of the class of transition matrices that the model admits?

[1]: Unlocking State-Tracking in Linear RNNs Through Negative Eigenvalues. Riccardo Grazzi, Julien Siems, Jörg K.H. Franke, Arber Zela, Frank Hutter, Massimiliano Pontil

---

> ### Author Rebuttal · Authors · 2025-03-31
>
> We thank the Reviewer for the positive perception. Below we try to answer to the reviewer’s remaining questions.
>
> __Q1: Can authors provide a wall-clock time analysis for their method against vanilla Mamba-2__
> Thanks. We will add the following to the paper comparing throughput and wall clock time (WCT) relative to the explicit model for the largest models (760M and 1.3B).
> ### Training Token Throughput and Wall Clock Time
> |Model|760M Tok/s|760M Rel T-put|760M Rel Time|1.3B Tok/s|1.3B Rel T-put|1.3B Rel Time|
> |-|-|-|-|-|-|-|
> |Mamba2*|1872|-|-|588|-|-|
> |Mamba2(4+1)|914|49%|205%|309|53%|191%|
> |Mamba2(24+4)|209|11%|180%|71|12%|166%|
> |ImpMambaAvg|546|29%|343%|184|31%|319%|
> |Llama†|775|-|-|582|-|-|
> |Llama(4+1)|472|61%|164%|237|41%|196%|
> |Llama(32+4)|47|6%|166%|50|9%|234%|
> |ImpLlamaAvg|131|16.8%|297%|149|26%|391%|
> The averaged numbers for the implicit models take the curriculum into account.
>
> ## Inference WCT Measurements (Time per token in milliseconds, averaged over 2048 tokens generated)
> |Steps|Llama 130M|Llama 1.3B|Mamba 130M|Mamba 1.3B|
> |-|-|-|-|-|
> |expl|46.7|97.9|23.5|45.7|
> |1|62.6|120.7|29.2|55.6|
> |2|130.8|236.7|53.5|102.9|
> |4|200.3|421.6|93.6|182.0|
> |8|440.3|745.2|180.2|3612.0|
> |16|748.8|1294.0|356.1|705.0|
> |32|1356.3|3204.8|710.5|1414.7|
>
> ## Memory Usage in Inference [MB] (Implicit / Explicit)
> |Model|Llama 130M|Llama 1.3B|Mamba 130M|Mamba 1.3B|
> |-|-|-|-|-|
> |Implicit / Explicit|871 / 511|10216 / 5281|935 / 547|10592 / 5488|
>
> ## Word Problem Inference
> We evaluate 4k samples at batch size of 512. For the single layer implicit Mamba2
> - setting an upper limit of 4 self-iterations takes 26s and gets 97.8% accuracy on the p=0.5 distribution
> - setting an upper limit of 16 self-iterations takes 35s and gets 99.8%. Note that iterations terminate after 6 steps on average due to convergence.
>
> The explicit model with 16 layers takes 37s and gets 1.5%. This explicit model has the same dimensions per layer, and hence 16x the number of parameters compared to the implicit model.
>
> ## Catbabi WCT Training
> |Model|GPU-Hrs|Rel T-put|Rel Time|
> |-|-|-|-|
> |expl-1-lyr|1.02|-|-|
> |impl-1-lyr|1.83|55.74%|179.41%|
> |expl-2-lyr|1.81|-|-|
> |impl-2-lyr|2.96|61.15%|163.54%|
> |expl-3-lyr|2.58|-|-|
> |impl-3-lyr|4.51|57.21%|174.81%|
>
> ## Catbabi WCT Inference (Time per token in milliseconds, averaged over 50 tokens generated)
>
> |Model|Step-1|Step-2|Step-4|Step-8|Step-16|Step-32|
> |-|-|-|-|-|-|-|
> |expl-1-lyr|1.879||||||
> |impl-1-lyr|2.966|4.8321|8.6203|16.0183|30.9154|60.5825|
> |expl-2-lyr|3.1831||||||
> |impl-2-lyr|4.7185|8.2655|14.8508|27.6253|51.3367|105.8571|
> |expl-3-lyr|4.4807||||||
> |impl-3-lyr|6.3261|11.0607|20.2967|38.7703|75.5918|149.1918|
>
> __Q2:  Could authors do a language modelling experiment with a control on the FLOPs.__
>
> Our primary goal is not FLOPs efficiency, but rather exploring a fundamental trade-off: how much true recursion is necessary for language modeling and reasoning, balancing parallelizability and expressiveness. While a single iteration of our implicit model approximately matches the explicit model in FLOPs, our intention isn't to claim FLOPs optimality. Instead, we aim to investigate qualitative differences, accepting increased representational power at the expense of greater and dynamic depth. Below, we provide a test-time compute table which allows for a FLOPs-matched comparison and highlights the flexibility of implicit models to trade off compute vs performance at test time.
>
> | Task    | Model                | f:1  | f:2  | f:4  | f:8  | f:16 | fixpt |
> |---------|----------------------|------|------|------|------|------|-------|
> | Avg Acc over tasks in Table1 | Implicit Mamba 1.3B|0.31|0.38|0.52|0.56|0.56|0.56|
> | Avg Acc over tasks in Table1 | Implicit Llama 1.3B|0.30|0.30|0.42|0.57|0.59|0.59|
>
>
> __Q3: Could the authors comment and contrast their method against SSM with complex diagonal values/negative eigen values [1]__?
>
> We discuss the paper that the Reviewer is referencing in the Related Work section. It is important to see that even with negative eigenvalues/complex diagonal values, the models in that reference are not able to solve the full S5 problem but only S5 restricted to transitions of (two-element) swaps (see Fig. 4 in Grazzi et al.).
>
> __Q4: Is it possible to get a characterization of the class of transition matrices that implicit model admits__?
>
> While a full characterization of the transition matrices goes beyond our study, Eq. (14) provides first insights. $\frac{\partial \varphi}{\partial h_{t-1}^*}$, the derivative of the fixed point w.r.t. hidden state in Eq. (12), is a source of general non-diagonal entries that depends on the implicit function $\varphi$. The hidden state is propagated through fully connected layers in the forward pass during the self-iteration. This leads to non-linear and non-diagonal contributions comparable with RNNs or multi-layer feed-forward networks.

---

### Official Review · Reviewer_gbY7 · 2025-03-17

**Overall Recommendation:** 2

**Summary:**

This paper proposes implicit language models, which are RNNs defined implicitly via fixed-point iterations. Theoretically, the authors show that implicit models can represent non-linear and non-diagonal state transitions of RNNs, overcoming the limitations of transformers and state-space models (SSMs) which are restricted to simpler, linear transitions. Empirical results show that implicit models can solve a challenging state-tracking problem that transformers and SSMs fail to learn. In addition, the authors scale implicit models up to 1.3B parameters and show they outperform explicit counterparts on language modeling benchmarks, with favorable length generalization and auto-regressive generation capabilities.

**Claims And Evidence:**

The study provides sufficient evidence to support its main theoretical and empirical claims:
- The claim that implicit state-space models can represent non-linear and non-diagonal state transitions is proven theoretically in Sec 3.1 and Appendix B.
- The authors reproduce the finding from prior work regarding the inability of transformers or SSMs to solve hard state tracking problems such as S5 and show that implicit SSMs can behave like RNNs, in Sec 4 and Fig 1.
- The scaling of implicit models to large language modeling tasks up to 1.3B parameters is supported by the results presented in Section 5 and the detailed experimental setup in Appendix D.3 (9 tasks).
- The claims regarding length extrapolation capabilities and duality between sequential and simultaneous modes are supported in Section 5, and Figs 4 and 2.

**Essential References Not Discussed:**

There are prior works that used simple and more advanced recurrent attention mechanisms based on the kernel-based view of attention for transformers (Tay et al. 2020). With such formulations of attention the transformers are converted into RNNs which do not have the problems as pointed out in this paper. It would be essential to discuss the unique advantages of implicit models beyond implementing non-linear state-transitions of RNNs which has been addressed in the past.

**Experimental Designs Or Analyses:**

Yes, I reviewed the experimental designs and analyses presented in the paper and found them to be sound and well-executed in general. Below I list a few non-major issues:

1. State tracking experiments
    - It would be useful to provide more details on how exactly the synthetic data distributions were created exactly and what is the intuition behind the chosen parameters. Providing some examples would also help.
    - In addition, a few experimental details are missing on the hyper-parameters of Mamba2 model on both synthetic and CATBABI tasks (number of layer, learning rates, batch sizes, etc).
2. Language modeling experiments
   - The scaling experiments would provide more confidence for practical impact if the mode size was up to at least 7B models; it is not guaranteed that the behavior observed below 1.3B will generalize and whether non-linear transitions are still useful.
    - It would help if experiments included more recent benchmarks for large language models such as MMLU, BBH, HELM.
    - Provide more details on the training budget used for training the implicit models and state-space models and quantify the training + inference costs. It was not clear to me how the authors ensure equal budget for convergence and what is the exact computational benefit for implicit models.

**Methods And Evaluation Criteria:**

- The proposed method is well motivated and shown theoretically to address the previous models' inability to represent non-linear state interactions. The expectation is that overcoming this limitation it will lead to more expressive modeling and state tracking.
- For evaluation, the authors focus on synthetic and real-world state tracking problems to systematically evaluate their theoretical findings. This is suitable experiment choice as it grounds the theoretical findings to empirical evidence, making the findings more trustworthy.
- In addition, the evaluate performance on downstream performance carrying out experiments with language models of increasing size up to 1.3B. These results show whether the expressivity is actually useful on language modeling and downstream real-world tasks.

**Other Comments Or Suggestions:**

N/A

**Other Strengths And Weaknesses:**

Strengths:
- The idea of using fixed-point iterations to combine the expressive power of RNNs and the parallelization benefits of transformers is quite interesting and useful, since the training becomes more efficient due to the computation of gradients with a constant memory footprint.
- Provides a convincing theoretical analysis of implicit models and makes a solid connection between the benefits of RNNs and Transformers.
- Experimentation is thorough and shows promising results up to 1.3B parameter models. The evaluation covers both tasks that require state tracking and tasks used in large language modeling studies.

Weaknesses:
- There is lack of discussion and comparison to hybrid models that leverage the benefits of RNNs and Transformer models through a combination of recurrent and softmax attention mechanisms.
- The family of implicit models provides an appealing solution for the lack of non-linear state transitions, however, the paper fails to motivate why that is useful in practical real-world tasks where instruction-following models based on Transformers perform exceptionally well.
- The scaling results up to 1.3B parameters do not provide concluding evidence that the state expressivity is actually needed for good performance on downstream tasks. In addition, the performance improvements also are not very consistent across different model sizes which further casts doubt on the generalizability to larger model sizes.

**Questions For Authors:**

1. Could you elaborate on how the implicit model are trained exactly? A more detailed explanation of the training details (e.g. masking, next-token prediction, etc) would help the reader better understand how this approach is implemented and what is the training cost involved.
2. Regarding the curriculum-based approach and the duration of different phases, what are the key considerations for the different design choices for different datasets? It would be good to specify how sensitive is the final performance on these choices.
3. Previous studies have devised hybrid architectures that leverage the benefits of both RNNs and Transformers. Are the expressivity issues applicable to them? How do implicit models compare to them?
4. What are the unique contributions of this work in the area of adaptive computation? Reading through the related work, it is not very clear what is the contribution of this work.
5. The implicit models are shown to have good properties but there is little emphasis on their limitations. I wonder if the authors could provide some additional discussion and analysis on which tasks or settings the implicit models have difficulties or underperform other models. Also, do the authors expect their results to generalize to even larger model sizes?

**Relation To Broader Scientific Literature:**

The contributions are generally well-situated within the broader scientific literature:
- The paper builds upon previous theoretical work that has identified limitations of state-space models in capturing complex sequential states and recognizing certain formal languages (Merill et al. 2024, Sarrof et al. 2024). They proposed models aim to address these exact limitations.
- The authors develop implicit models building on top of deep equilibrium models and implicit function theorem from prior work. The adaptive computation that is inherited from these models is a useful property that has shown to be useful in previous research (Graves 2017, Dehghani et al. 2019).
- Provides additional evidence to the existing literature that looped models are able to generalize better to input lengths not seen during training (Yang et al. 2024a).

It would be useful to discuss the advantages of implicit models compared to recurrent-attention-based transformers or hybrid transformers that make use of full and recurrent attention mechanisms that are competitive in terms of quality and speed tradeoff.

**Theoretical Claims:**

The main theoretical claim of the paper is captured in Theorem 1 and shows that the transition function defined by the implicit SSM is non-linear and non-diagonal. I checked the proof and found it to be logically correct; it applies the implicit function theorem to the function $g(z, h, x, \theta) = z - f_\theta(z, h, x)$ and then shows that the derivative of the implicit function $\phi(h,x, \theta)$ with respect to $h$ is  a non-linear function if $f$ is non-linear. Based on this then the state-to-state Jacobian is shown to be non-linear.

---

> ### Author Rebuttal · Authors · 2025-03-31
>
> We thank the reviewer for their constructive feedback and positive assessment of our submission. Below we address raised concerns and assumptions:
>
> **Experimental details**: we will make sure to double check Appendix D to see if any detail is missing. Please see additional wall clock time and memory footprint in response to reviewer __AqCH__.  We are also releasing code with precise experiment configurations.
>
> **Essential references not discussed:** The Reviewer appears to make the key assumption that some or all of the models with linear attention or kernelized attention as discussed in (Tay et al. 2020), or models with gated linear attention and variations of gated state-space models such as Mamba (Gu & Dao 2024) already address the theoretical issues discussed in our manuscript. We would like to respectfully clarify this assumption. Yes, these models admit a recurrent inference mode. [1] discusses the structural similarities between these attention variants. However, they all face the same limitations discussed in (Merill et al. 2024, Sarrof et al. 2024) since their recurrence is not expressive enough. We discuss xLSTM as an exception, whose sLSTM module is a non-linear + non-diagonal RNN, which lacks parallelization, though. If we should be missing relevant related work, we are happy to take further suggestions.
>
> [1] https://arxiv.org/abs/2405.15731
>
> **Practical usefulness:** Our primary goal is to highlight qualitative differences between implicit and explicit sequence models, without claiming universal superiority. Our experiments demonstrate that implicit models scale practically, as further confirmed by concurrent work (https://arxiv.org/pdf/2502.05171). Thus, implicit models are a viable choice when complex sequential processing (e.g., state tracking) is required. We offer a design point where you can get a qualitative jump in expressiveness at the expense of compute, a trade-off reminiscent of the test-time compute paradigm.
>
> > [...] however, the paper fails to motivate why that is useful.
>
> Static depth models show exceptional performance. Yet, they even struggle with certain regular languages. This limits their capabilities to execute tasks that require state tracking (e.g. managing supply chain processes). We tested GPT 4o and o1 (API) on the S5 state-tracking task.
> |Sample Length|o1-mini Acc|GPT-4 Acc|
> |-|-|-|
> |5|0.967|0.200|
> |15|0.967|0.100|
> |32|0.100|0.133|
>
> > 1.3B parameters do not provide concluding evidence that the state expressivity is actually needed for good performance
>
> We agree that most downstream tasks do not require extensive state tracking, and we do not claim that it is required in all tasks. We do see increased performance in HellaSwag, a benchmark requiring limited state-tracking.
>
> # Questions
>
> 1.Implicit models for language modeling and CatbAbI used next-token prediction loss with phantom gradients where backpropagating through fixed self-iterations (4 for language modeling, 6 for CatbAbI) after gradient-free searches (up to 24/32 iterations language modeling, 50 CatbAbI).
>
> 2.The LM (The Pile) curriculum balances cost and accuracy by applying more self-iterations to only n=20% of tokens (Table 1). To address Reviewer questions, we tested n=10% on the 1.3B Mamba2 (24+4) model, showing curriculum robustness.
>
> |Metric|10%|20%|
> |-|-|-|
> |Tokens seen|207|207|
> |LAMBADA |0.4186|0.4116|
> |HellaSwag|0.3672|0.3527|
> |PIQA|0.6502|0.6572|
> |Arc-E|0.4596|0.4815|
> |Arc-C|0.2372|0.2372|
> |Wino|0.5170|0.5130|
> |OpenQA|0.2900|0.3000|
> |Average|0.4200|0.4219|
>
> 3.See initial statement
>
> 4.Implicit models can implement non-diagonal + non-linear transitions. We show in Fig. 2 (and Sec. 5) that implicit models allow to conduct sequential inference, e.g. language generation, only carrying forward the converged hidden state (SSM) or KV-cache (Transformer). This allows memory allocation independent of the number of iterations. While RNN-based ACT (Graves, 2017) make the non-parallizability of RNNs even worse by sequentially iterating more steps per token, implicit models learn the adaptive budget for all tokens in parallel, which allows us to scale to 1.3B models.
>
> 5.We will further discuss limitations and computational differences between implicit and explicit models. Larger models suffer less from state tracking (Fig. 1), but even the largest GPT models remain limited.
>
> # New experiments
> > "Benchmarks"
>
> We were able to evaluate the models on an additional task, the MMLU. Please note that we selected our benchmarks to align with the Mamba2 and xLSTM papers.
>
> |Model|MMLU Accuracy|
> |-|-|
> |1.3B_implicit_mamba2 (24+4)|0.269|
> |1.3B_implicit_llama (32+4)|0.258|
> |1.3B_llama_baseline|0.2483|
> |1.3B_mamba_baseline|0.2502|
>
> **Computational benefits:**
> Note that implicit models have qualitatively higher expressivity than explicit models as shown in our synthetic experiments. In addition, we show results depending on the compute budget in our response to __AqCH__.

---

### Decision · Program_Chairs · 2025-05-01

**Decision:**

Accept (spotlight poster)

**Comment:**

The authors propose implicit language models, RNNs defined implicitly via fixed point iteration. They show theoretically that this architectures can simulate non-linear RNNs, therefore from a representational point of view they can represent functions SSM cannot. Particularly in the rebuttal the theoretical statement is made a bit more sharp. The authors also show empirical evidence for the performance of their proposed model, up to size 1.3B.

I think this work is a step forward towards better understanding recurrent models and also making them a valid contender to transformers. I believe the authors have provided a very detailed rebuttal, answering most of the question raised, and have improved their paper in the process (including improving the theoretical result by making it more sharp). The empirical evaluation is also thorough and provides a clear signal that this direction is worth pursuing. Overall I think this is a strong contribution to the field.
I do urge the authors to include all provided details in the rebuttal into the final paper, since I think the rebuttal made the paper stronger.